# Parent of origin genetic effects on methylation in humans are common and influence complex trait variation

Yanni Zeng[1], Carmen Amador [1], Charley Xia[1,2], Riccardo Marioni[3,4], Duncan Sproul [1,5], Rosie M. Walker[3,4], Stewart W. Morris[4], Andrew Bretherick[1], Oriol Canela-Xandri[1,2], Thibaud S. Boutin [1], David W. Clark[6], Archie Campbell [4], Konrad Rawlik [2], Caroline Hayward [1], Reka Nagy[1], Albert Tenesa [1,2], David J. Porteous [3,4], James F. Wilson [1,6], Ian J. Deary[3,7], Kathryn L. Evans[3,4], Andrew M. McIntosh [3,8], Pau Navarro [1] & Chris S. Haley [1,2]

Parent-of-origin effects (POE) exist when there is differential expression of alleles inherited from the two parents. A genome-wide scan for POE on DNA methylation at 639,238 CpGs in 5,101 individuals identifies 733 independent methylation CpGs potentially influenced by POE at a false discovery rate ≤ 0.05 of which 331 had not previously been identified. *Cis* and *trans* methylation quantitative trait loci (mQTL) regulate methylation variation through POE at 54% (399/733) of the identified POE-influenced CpGs. The combined results provide strong evidence for previously unidentified POE-influenced CpGs at 171 independent loci. Methylation variation at 14 of the POE-influenced CpGs is associated with multiple metabolic traits. A phenome-wide association analysis using the POE mQTL SNPs identifies a previously unidentified imprinted locus associated with waist circumference. These results provide a high resolution population-level map for POE on DNA methylation sites, their local and distant regulators and potential consequences for complex traits.

[1] MRC Human Genetic Unit, Institute of Genetics and Molecular Medicine, University of Edinburgh, Edinburgh EH4 2XU, UK. [2] The Roslin Institute and Royal (Dick) School of Veterinary Sciences, University of Edinburgh, Edinburgh EH25 9RG, UK. [3] Centre for Cognitive Ageing and Cognitive Epidemiology, University of Edinburgh, Edinburgh EH8 9JZ, UK. [4] Centre for Genomic and Experimental Medicine, IGMM, University of Edinburgh, Edinburgh EH4 2XU, UK. [5] Edinburgh Cancer Research Centre, Institute of Genetics and Molecular Medicine, University of Edinburgh, Edinburgh EH4 2XR, UK. [6] Centre for Global Health Research, Usher Institute of Population Health Sciences and Informatics, University of Edinburgh, Edinburgh EH8 9AG, UK. [7] Department of Psychology, University of Edinburgh, Edinburgh EH8 9JZ, UK. [8] Division of Psychiatry, University of Edinburgh, Edinburgh EH10 5HF, UK. Correspondence and requests for materials should be addressed to C.S.H. (email: chris.haley@igmm.ed.ac.uk)

DNA methylation plays a crucial role in regulating gene expression and maintaining genomic stability[1]. Inter-individual variation of DNA methylation levels at CpG sites (henceforth methylation CpGs) has been associated with complex diseases, quantitative traits, environmental exposures and the aging process[2–6]. Previous studies have estimated that on average across sites, 19% of variation in DNA methylation level is contributed by the additive genetic effects[7]. A number of genetic variants have been shown to regulate methylation CpGs (such variants being termed methylation quantitative trait loci (mQTL)) in an additive manner, acting locally (*cis*) or distantly (*trans*)[8,9]. In contrast, shared family environment has been shown to have a relatively smaller overall contribution (3% on average across sites) to variation in DNA methylation[7,10].

DNA methylation can also be affected by parent-of-origin effects (POEs), which are non-additive genetic effects that manifest as phenotypic differences depending on the allelic parent-of-origin[11]. There are several possible causes of observed POEs, but the most common is genomic imprinting[11]. These POEs caused by imprinting can lead to different phenotypic patterns, including classical paternal or maternal imprinting, and other complex forms (Supplementary Fig. 1)[11]. (We use the standard definition of maternal imprinting, where the maternal allele is silenced and the paternal expressed and vice versa for paternal imprinting.) Although previous studies estimated the number of imprinted expressed genes in the human genome at around 100[12], POEs caused by imprinting can spread to wider genomic regions through regulatory variants located in imprinted regions transmitting the POEs to their genomic targets (Supplementary Fig. 1)[13,14]. POEs caused by imprinting have been detected at the DNA methylation, gene expression and phenotypic levels[13–15]. The precise regulation of expression of genes influenced by imprinting is crucial for embryonic development, metabolism and behavioural traits, and the effect can last into later life[16–18]. Given the regulatory role of DNA methylation on gene expression, the identification of POEs on DNA methylation is of particular importance to facilitate understanding of the molecular mechanism of the POEs observed at the gene expression or phenotypic level[16].

Recent progress has been made towards profiling genome-wide imprinted regions and POE caused by imprinting in DNA methylation[13,19–21]. Imprinted regions can be identified by bisulfite sequencing detecting CpGs that display imbalanced or bimodal methylation between paired chromosomes, which could be caused by imprinting[19,21]. POEs in these regions can be explicitly modelled in association tests between SNPs and methylation levels for each CpG, assuming allelic effects differ between maternally and paternally inherited alleles[13,14]. Whereas this approach enables both the identification of imprinting-associated POE in methylation levels and the localisation of the SNPs associated with that POE, the limitation lies in the huge multiple testing burden introduced by the number of SNP–CpG pairwise tests for a genome-wide scan. Furthermore, the localisation of POEs from individual SNP does not ensure the elucidation of the overall genetic architecture underlying methylation levels at each CpG, particularly when multiple POEs from multiple independent SNPs target the same CpG. Therefore, further methodological advances are required in order to improve understanding of the role of POEs in the genetic control of methylation levels and hence potentially better explain the influence of POEs on phenotypic variation.

Imprinting-caused POEs on DNA methylation may have downstream effects on complex traits. Others have shown that applying models which account for POEs in genome-wide association studies (GWAS) can identify genetic variants that underlie POEs on multiple complex traits and diseases[18,20,22], and that

SNPs which play a regulatory role in DNA methylation through POEs also have significant associations in GWAS performed using an additive model[13]. Combining these with previous observations that disease-associated loci are enriched in regulatory regions[23], analyses that link POE regulation to DNA methylation and POE regulation to complex traits potentially provide important insights for the understanding of both non-additive genetic and epigenetic control mechanisms for complex traits and diseases.

We propose here a variance component method to detect signatures of POEs caused by imprinting in the human DNA methylome, by identifying methylation CpGs showing an unusually increased full-sibling and/or one-parent–offspring similarity. Using this method, we perform a genome-wide scan for POE on each of 639,238 methylation CpGs in a homogenous Scottish sample ($N = 5101$) with complex pedigree structure[24], in which both previously unknown and known POE-influenced CpG sites are identified. We then perform a POE–mQTL analysis to identify local and distant regulatory genetic variants of methylation at the POE-influenced CpG sites identified in the variance component analyses. This is followed by an analysis to identify complex traits associated with the detected POE-influenced methylation CpGs. We also use identified POE–mQTL SNPs to guide a phenome-wide association analysis, through which we identify one locus affecting waist circumference through a previously unidentified POE, demonstrating that the use of methylation data and the proposed set of analyses contribute to increase our understanding of the non-additive genetic control of complex traits.

## Results

**Overview of the study design**. Table 1 shows a summary of the study design. An established five-component variance component analysis accounting for genetic and environmental variation was first used to partition DNA methylation variation for each measured CpG. Following this, a two-stage pipeline was applied to identify potential POE-influenced methylation CpGs among all measured CpGs. The first stage applied a POE variance component method that targeted the localisation of POE-influenced methylation CpGs. The second stage applied a POE–mQTL (parent-of-origin effect mQTL) analysis that accounted for POE to localise the SNPs that introduce the POE on the identified CpG candidates from the first stage. This was followed by two additional analyses to profile the phenotypic consequence of the POE-influenced methylation CpGs and their POE–mQTL SNPs on complex traits.

**Genetic and environmental contributions to DNA methylation**. Using the GKFSC variance component model, we decomposed methylation variation at 639,238 CpGs into contributions from two genetic and three family environmental effects, including the additive genetic effect of common SNPs ($h_g^2$), an additional additive genetic effect associated with pedigree ($h_k^2$), shared environmental effects between nuclear family members ($e_f^2$), shared environmental effects between full siblings ($e_s^2$) and shared environmental effects between members of couples ($e_c^2$). The additive genetic effect ($h_g^2 + h_k^2$) was the largest contributor (Table 2), explaining an average of 16.7% of the variation in DNA methylation (this average includes sites for which the additive genetic effect does not explain any variation), with an estimate of 9.5% and 7.2% of the DNA methylation variation explained by the common SNP-associated and the pedigree-associated additive genetic components, respectively. The contribution from common SNPs varied across genomic regions, with an increased

**Table 1 Study design**

| ANALYSIS | AIM | MODEL | $N_{TESTS}$ | $N_{SIGRESULTS}$ |
|---|---|---|---|---|
| GKFSC VC | Understand sources of variation of methylation at CpG sites | CpG ~ G + K + F + S + C | 639,238 (CpGs) | G: 24,101<br>K: 1531<br>F: 0<br>S: 78<br>C: 0 |
| POE-targeted VC | Find POE-influenced CpGs | Base: CpG ~ G + K<br>Complex: CpG ~ G + K + S<br>Maternal: CpG ~ G + K + $S_M$<br>Paternal: CpG ~ G + K + $S_P$ | 639,238 (CpGs) | Complex: 606<br>Maternal: 220<br>Paternal: 158<br>Total: 984 |
| POE–mQTL | (a) Find POE-influenced CpGs<br>(b) Find SNPs associated with POE-influenced CpGs | CpG ~ $SNP_{ADD}$ + $SNP_{DOM}$ + $SNP_{POE}$ | 7e9 (984 CpGs*7e6 SNP) | CpGs: 586<br>POE–mQTLs:<br>*cis*: 1814<br>*trans*: 103 |
| POE–EWAS | Phenotypic consequence of POE-influenced CpGs | Trait~CpG | 26,568 (984 CpGs*27 independent traits) | CpGs: 14<br>Traits: 10 |
| POE–PheWAS | Phenotypic consequence of POE–mQTL SNPs | Trait~$SNP_{ADD}$ + $SNP_{DOM}$ + $SNP_{POE}$ | 51,165 (1895 independent mQTLs*27 independent traits) | Traits: 1<br>SNPs: 1 |

The table shows an overview of the analyses performed (ANALYSIS), describing their aims (AIM) and the models used (MODEL), as well as the number of tests performed ($N_{tests}$) and the number of significant results obtained ($N_{SIGRESULTS}$)

*GKFSC VC* variance component analyses to partition methylation level variation into its additive genetic (G: SNP associated, K: pedigree associated) and non-additive/environmental (F: family, S: sibling, C: couple) components, *SNP* single-nucleotide polymorphism

*POE-targeted VC* modified variance component analysis detects candidate methylation sites with parent-of-origin inheritance pattern (parent-of-origin effect, POE). Base: model without POE; complex: model including a complex POE component allowing for increased similarity between siblings; maternal: model including a POE component ($S_M$) allowing for increased similarity between father and offspring and siblings; paternal: model including a POE component (Sp) allowing for increased similarity between mother and offspring and siblings;

*POE–mQTL* parent-of-origin effect methylation quantitative trait loci analyses, *ADD* Additive effect, *DOM* dominance effect,

*POE-EWAS*, complex trait association with methylation levels of POE CpGs,

*POE-PheWAS*, phenotype-wide association study accounting for parent-of-origin effects for parent-of-origin methylation level associated loci (POE–mQTL)

**Table 2 DNA methylation variation decomposed into genetic and environmental components**

| Source | Mean PV | Maximum PV | First quartile PV | Third quartile PV | Nominal Sig. sites | Genome-wide Sig. sites |
|---|---|---|---|---|---|---|
| **G** | 9.5% | 99.20% | 0.67% | 12.98% | 162,800 | 24,101 |
| **K** | 7.2% | 97.10% | 0.00% | 11.05% | 59,117 | 1531 |
| **F** | 1.2% | 19.10% | 0.00% | 1.84% | 1946 | 0 |
| **S** | 1.4% | 46.30% | 0.00% | 2.24% | 23,600 | 78 |
| **C** | 2.1% | 33.50% | 0.00% | 3.18% | 14,514 | 0 |

Proportion of variation in methylation levels at the 639,238 studied CpG sites explained (PV) by **G** (common SNP-associated additive genetic component), **K** (pedigree-associated additive genetic component), **F** (shared environmental effects between nuclear family members), **S** (non-additive genetic or shared environmental effects between full siblings) and **C** (shared environmental effects between members of a couple). The number of CpG sites that were significant in the component of interest, both at nominal and genome-wide level (Sig. sites at nominal and genome-wide levels) is also shown for each of the five components fitted

contribution for CpGs within enhancer regions ($h_g^2 = 13\%$ for CpGs in enhancers vs. $h_g^2 = 9\%$ for CpGs outside them), and a decreased contribution for CpGs surrounding TSSs (Supplementary Fig. 2). Shared environmental effects contribute an average of 1.2–2.1% of the variation in DNA methylation (Table 2), but the contributions also vary across genomic regions (Supplementary Fig. 3).

The number of CpGs with a statistically significant proportion of methylation variance explained by G, K, F, S or C, in the GKFSC model was 24101, 1531, 0, 78 and 0, respectively (Table 2). CpGs that showed genome-wide significance for the full-sibling component $e_s^2$ (N=78) were in regions highly enriched in published genomic imprinting regions (with 58 of the 78 CpGs being located within 2 kb of known imprinted regions, $P_{(Fisher\ exact\ test)} = 1.3 \times 10^{-80}$), suggesting that (1) POEs caused by imprinting are likely to contribute to the variation of a subset of CpGs, (2) besides any shared environmental effect, the full-sibling associated component ($e_s^2$) also captures POE caused by imprinting (for a more detailed discussion see the Methods section) and (3) variance component analysis that accounts for the increased similarity between full siblings can be applied to identify CpGs potentially influenced by POEs caused by imprinting (see below).

**POE-targeted variance component analysis.** We developed a model selection-based approach to perform a genome-wide scan to identify methylation CpGs potentially influenced by POEs caused by imprinting targeting three main patterns of imprinting: paternal, maternal and complex (Fig. 1)[11]. Since each pattern reflects different phenotypic similarities between nuclear family members, for each CpG we tested three alternative models (complex, paternal and maternal), and performed model selection to select the best model for each CpG, that is, the model that better describes the observed phenotypic similarity (Aim 2 in Table 1).

This genome-wide scan identified 984 methylation CpGs that exceeded genome-wide significance for POEs at a FDR ≤ 0.05 level (Supplementary Fig. 4). Of these 984 POE-influenced CpG candidates, the selected model was complex imprinting for 606, paternal imprinting for 158 and maternal imprinting for 220 CpGs (Supplementary Data 1). An example of the genome-wide scan results for one of the previously unidentified maternal imprinting sites is shown in Fig. 2. The 984 CpGs included some in well-known imprinted genes, such as IGF2 and PEG3 (Supplementary Fig. 5, Supplementary Data 1), but more generally were located in genomic regions highly enriched in known imprinted regions, particularly when extending those

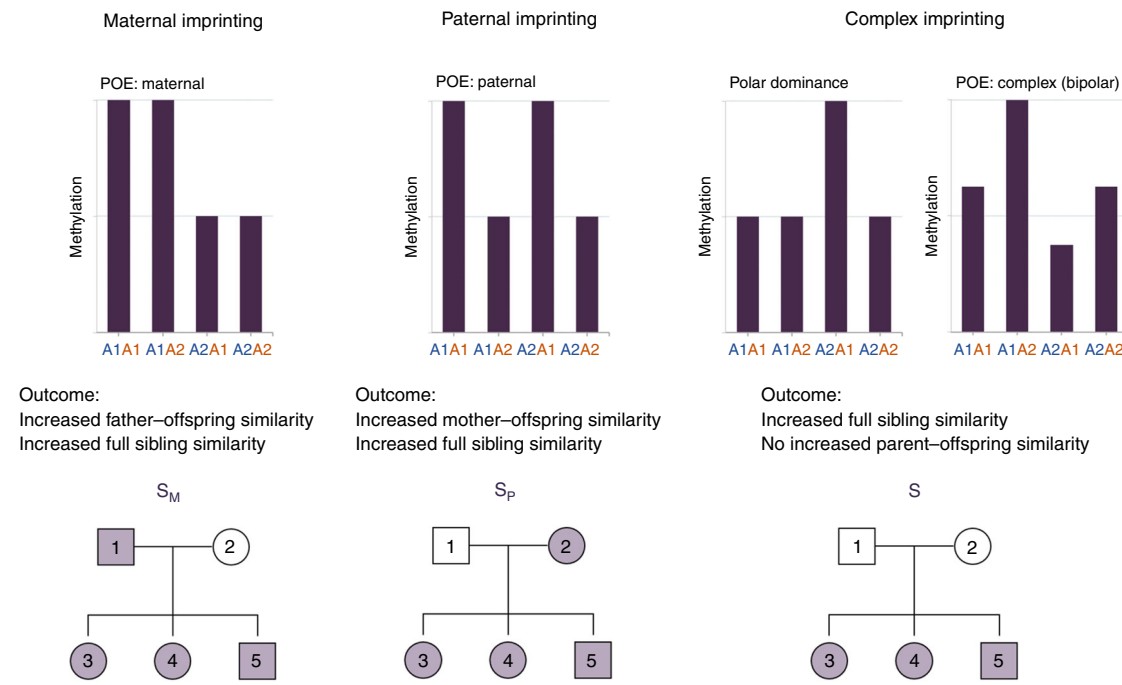

**Fig. 1** The expected phenotypic covariance structures between nuclear family members introduced by different POE patterns. The bar charts show putative levels of methylation associated with the four possible genotypes at a SNP controlling imprinting (paternal allele in blue, maternal allele in red). The family pedigrees show as shaded the family members between which similarity in methylation is increased due to these patterns of imprinting

known regions by 2 kb (OR = 15.3 (13.1–17.7), $P_{(Fisher\ exact\ test)} = 5.3 \times 10^{-171}$, Supplementary Table 1). When overlapping these 984 CpGs with regions of different chromatin states and subgenic structures, these CpGs were in regions enriched in noncoding RNA ($P_{(Fisher\ exact\ test)} = 1.05 \times 10^{-5}$), Polycomb repressed regions ($P_{(Fisher\ exact\ test)} = 1.59 \times 10^{-8}$), weak enhancers ($P_{(Fisher\ exact\ test)} = 6.57 \times 10^{-11}$) and were depleted in active promoter regions ($P_{(Fisher\ exact\ test)} = 2.11 \times 10^{-9}$) (Supplementary Data 2, Fig. 3, Supplementary Fig. 6). Compared with published epigenome-wide association studies (EWASs) (Supplementary Table 2), the 984 CpGs were also enriched in genic regions of genes containing methylation sites associated with body mass index (BMI) ($P_{(Fisher\ exact\ test)} = 4.85 \times 10^{-9}$)[3] and alcohol consumption ($P_{(Fisher\ exact\ test)} = 2.67 \times 10^{-6}$)[25] (Supplementary Data 2, Fig. 3). CpG sites were assigned to the nearest gene if they were located between 5 kb 5′ and 1 kb 3′ of the gene boundary. A gene set (pathway) analysis shows that the annotated genes were enriched in the Type I diabetes mellitus pathway ($P_{(EASE\ test)} = 4.92 \times 10^{-5}$) (Supplementary Data 3).

Clumping (see the Methods section) the 984 CpGs based on their methylation correlations resulted in 733 independent sites of which 331 were previously unidentified (Supplementary Data 1), as they are located more than 2 Mb away from previously reported imprinting-influenced regions[13].

**POE–mQTL analysis**. Imprinted genetic variants potentially underlie the observed POEs affecting methylation levels at the 984 candidate CpGs identified by the variance components analyses (Supplementary Fig. 1)[11]. To identify variants causing POEs on methylation CpGs, a POE–mQTL analysis was performed for each of the 984 CpGs (Table 1, Aim 3). We used genome-wide imputed common SNPs, and assigned alleles to a paternal or maternal origin for individuals with pedigree information. This information was used to model an additive, a dominant and a POE effect, and

these were fitted as explanatory variables for methylation levels at each CpG site (see the Methods section). This revealed that among the 984 CpGs (733 independent loci), 60% (586/984) of CpGs and 54% (399/733) of independent loci have at least one cis- or trans-POE–mQTL identified (Table 3, Supplementary Data 1, Supplementary Fig. 4); 58% (569/984) of CpGs have at least one cis-POE–mQTL (Supplementary Data 4), and 6.8% (67/984) have at least one trans-POE–mQTLs (Supplementary Data 5). For these 586 CpG sites, the identification of POE–mQTL SNPs provides strong evidence for the POE caused by imprinting (Table 3). A total of 1814 independent cis-POE–mQTLs were identified, 1% (18/1814) were also trans-POE–mQTLs, and 22% (409/1814) and 11% (202/1814) were previously identified as eQTLs and mQTLs, respectively, using additive genetic models[8,9,26,27]. Both cis- and trans-POE–mQTL SNPs were in regions highly enriched for known imprinted regions (cis: $P_{(Fisher\ exact\ test)} = 0$, OR = 7.8 (7.6–8.1); trans: $P_{(Fisher\ exact\ test)} = 2.0 \times 10^{-78}$, OR = 10.2 (8.4–12.2)), and non-genetic regulated imbalanced methylation regions as reported in ref. [19] (cis: $P_{(Fisher\ exact\ test)} = 0$, OR = 3.2 (3.1–3.4); trans: $P_{(Fisher\ exact\ test)} = 1.1 \times 10^{-61}$, OR = 6.5 (5.4–7.8)). They were enriched but to a lesser extent in previously defined[28,29] imprinting control regions (ICR) (cis: $P_{(Fisher\ exact\ test)} = 9.8e-293$, OR = 2.4 (2.5–2.6); trans: $P_{(Fisher\ exact\ test)} = 0.23$, OR = 1.3 (0.8–1.8)). For each independent CpG site with at least one POE–mQTL ($N_{total} = 586$, $N_{independent} = 399$), a median of six independent cis-POE–mQTLs and one independent trans-POE–mQTLs were identified. For CpGs showing POE with a maternal or paternal pattern as identified in the variance component analysis, we could infer the parental origin of the effective and silenced alleles in the POE–mQTL SNPs, based on the relative signs of the additive effect and the POE in the POE–mQTL model. This comparison could only be made when minor homozygotes at the SNP were present in the sample (so that the additive effect can be distinguished from the dominance effect) and the additive effect was significant ($P_{(t\ test)} \leq 0.001$). The results showed a consistency

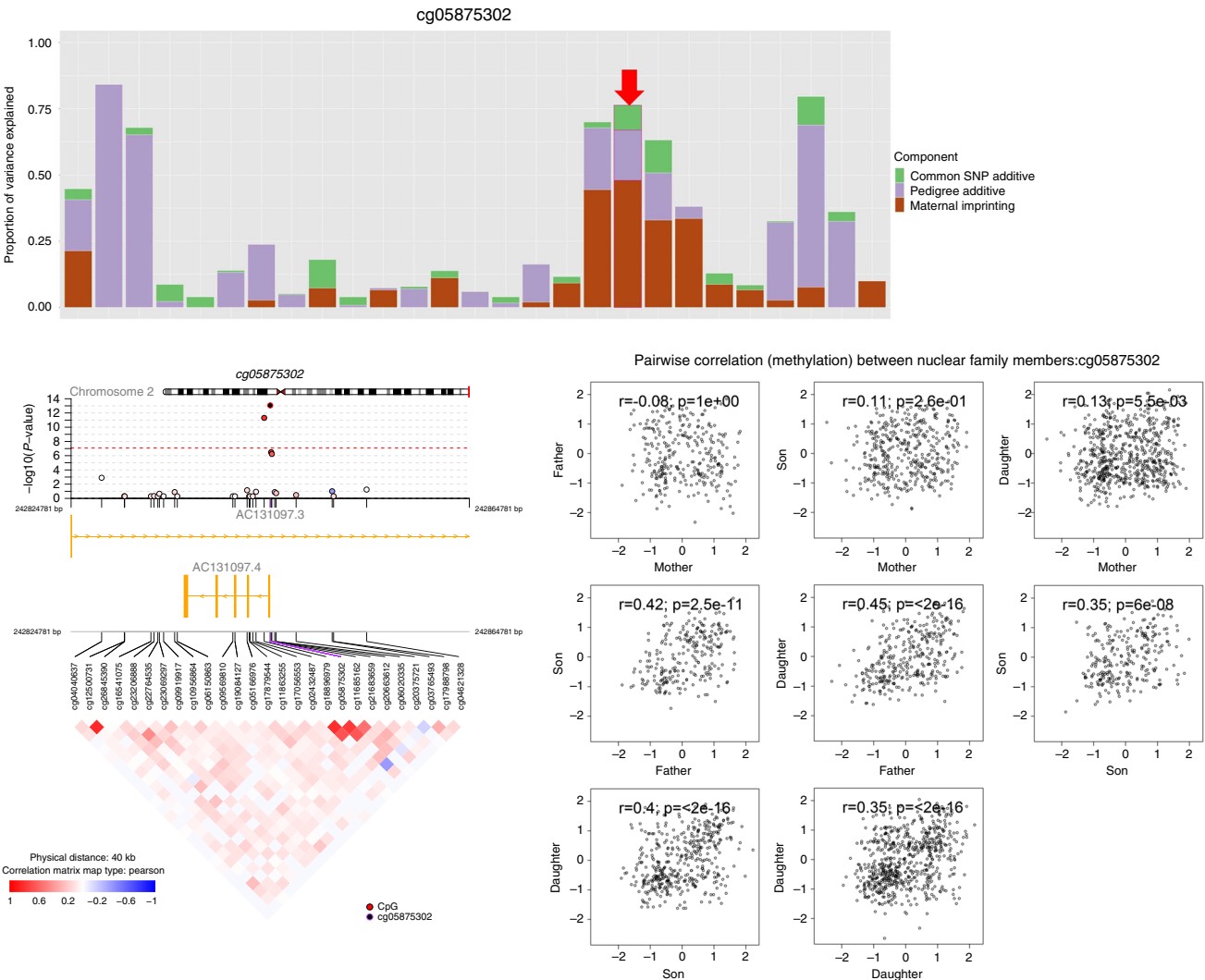

**Fig. 2** Example of a novel CpG site (cg05875302) influenced by maternal imprinting POE. Upper panel: bars represent estimated variance explained by each component in the selected model for the site displaying significant maternal imprinting (cg05875302, red arrow) and the sites within 20 kb on either side of the selected site. Bottom left panel: regional plot of −log$_e$ (p-value from LRT) of the POE in the selected (red ringed black dot) and surrounding CpG sites with matrix of pairwise correlations of methylation level between these sites in the heatmap below. Bottom right panel: pairwise correlation between methylation M values (corrected for technical and biological covariates) between different pairs of nuclear family members

of over 99% in the inference of the nature of parental effect (paternal or maternal) inferred from POE–mQTL analysis with that inferred from the variance component analysis.

Two independent *trans*-POE–mQTLs were identified as regulatory hubs, as they regulated more than one independent CpG target, both in *cis* and in *trans*. As an example, SNP rs231356 (chr11:2705343) was identified as a *cis*-POE–mQTL for three CpG sites (cg14958441, cg09518720 and cg02219360) on the same chromosome 11, the three displayed a complex bipolar imprinting pattern (Fig. 4, Supplementary Table 3). rs231356 also acted as a *trans*-POE–mQTL for another two CpGs, one on chromosome 18 (cg05884032) and one on chromosome 13 (cg23776532), both displaying a paternal imprinting pattern (Supplementary Table 3, Fig. 4). Notably, we failed to identify any *cis*-POE–mQTL for these two CpGs, suggesting that the *trans* effects from SNP rs231356 were potentially the cause of the parent-of-origin inheritance pattern detected in the variance component analyses for the two CpGs.

We compared our results with those published in a recent study, which also applies a POE–mQTL analysis on DNA

methylation data (437,542 CpG sites) measured longitudinally in blood on a smaller sample (N$_{offspring}$ = 740)[13]. Among the 327 CpGs that were associated with the POE from genetic variants (199 SNPs) detected[13], 260 CpGs were also analysed in our study. In our variance component analysis, 65% of those CpGs (162/260) showed a significant POE that exceeded the Bonferroni-corrected threshold for a replication and 50% of them (129/260) reached genome-wide significance (Supplementary Data 6). As we performed POE–mQTL analysis only for the 984 CpGs showing significant POE in our variance component analysis, we can only compare POE–mQTL results for the 129 CpGs analysed in both studies. We detected genome-wide significant POE–mQTL SNPs for all of the 129 CpGs in our cohort, and 94% (121/129) of those CpGs found an association with the same POE–mQTL SNPs reported[13].

To explore if candidate POE-influenced CpGs with at least one POE–mQTL SNP association (classified as strong POE evidence in Table 3, N = 586) differ from candidates without a POE–mQTL (classified as moderate POE evidence in Table 3, N = 398), we performed additional enrichment analysis for each

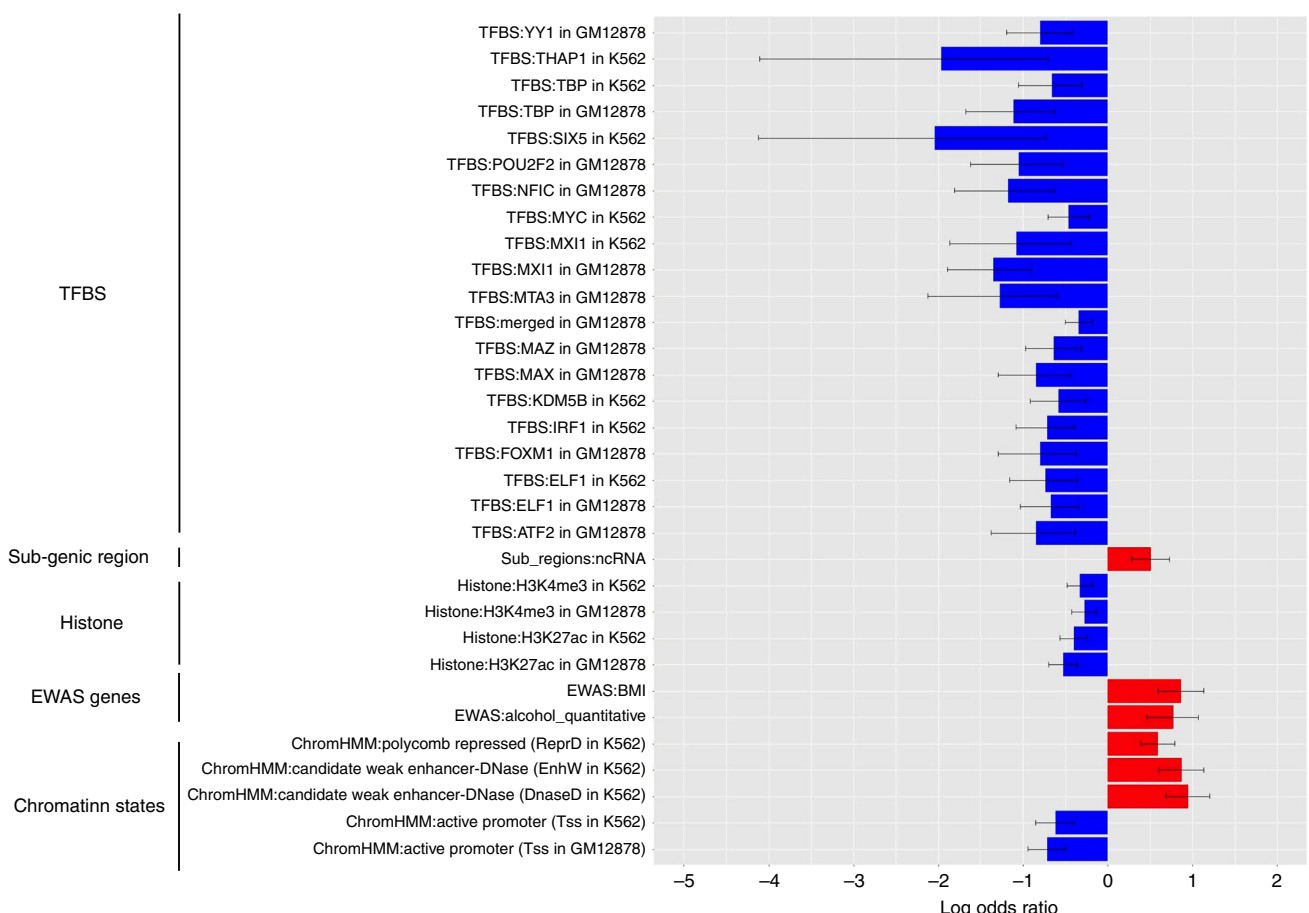

**Fig. 3** Genomic annotations significantly enriched in (red) or depleted of (blue) POE-influenced methylation CpGs. Error bars: 95% confidence interval

**Table 3 Classification of the identified 984 CpGs potentially influenced by POEs**

| Strength of Evidence | POE-mQTL | Other studies | $N_{CpG}$ | VC POE model | $N_{CpG}$ per POE model |
|---|---|---|---|---|---|
| Strong | √ | Replication (<2 kb) | 223 | Maternal/paternal | 104 |
| | | | | Complex | 119 |
| Strong | √ | Overlap (2 kb−2 Mb) | 172 | Maternal/paternal | 79 |
| | | | | Complex | 93 |
| Strong | √ | Not identified (>2 Mb) | 191 | Maternal/paternal | 70 |
| | | | | Complex | 121 |
| Total strong | | | 586 | | 586 |
| Moderate | × | Replication (<2 kb) | 11 | Maternal/paternal | 2 |
| | | | | Complex | 9 |
| Moderate | × | Overlap (2 kb–2 Mb) | 190 | Maternal/paternal | 71 |
| | | | | Complex | 119 |
| Moderate | × | Not identified (>2 Mb) | 197 | Maternal/paternal | 52 |
| | | | | Complex | 145 |
| Total moderate | | | 398 | | 398 |
| Total | | | 984 | | 984 |

The table classifies the 984 candidate CpG sites identified with the targeted POE variance component (VC) analysis into groups representing the support for the detected POE (strength of evidence) based on having or not an identified POE–mQTL (POE-mQTL), and if their position overlaps with previously published studies (other studies: replication: the CpG is in a region within 2 kb of a known imprinted region; overlap: the CpG is in a region between 2 kb and 2 Mb of a known imprinted region; not identified: the CpG is more than 2 Mb away from a known imprinted region). $N_{CpGs}$ is the number of CpG sites in each category. VC POE model indicates the selected VC model (maternal/paternal or complex imprinting) and $N_{CpGs}$ per POE model is the number of CpG sites in each subgroup

group separately (Supplementary Data 7, 8). The results showed that CpGs in both groups were in regions depleted in promoters and enriched in Polycomb-repressed regions (Supplementary Fig. 7). The stronger POE evidence group displayed a much higher enrichment in known imprinted regions (based on a 2 kb distance from the regions, $P_{(Fisher\ exact\ test)} = 3.2 \times 10^{-214}$), noncoding RNA ($P_{(Fisher\ exact\ test)} = 2.6 \times 10^{-11}$), genic regions of genes containing methylation sites associated with BMI ($P_{(Fisher\ exact\ test)} = 2.3 \times 10^{-9}$) and alcohol consumption ($P_{(Fisher\ exact\ test)} = 8 \times 10^{-6}$) (Supplementary Data 7), whereas the moderate POE evidence group showed enrichment in genic regions of genes containing methylation sites associated with smoking ($P_{(Fisher\ exact\ test)} = 2.8 \times 10^{-5}$) (Supplementary Data 8, Supplementary Fig. 7).

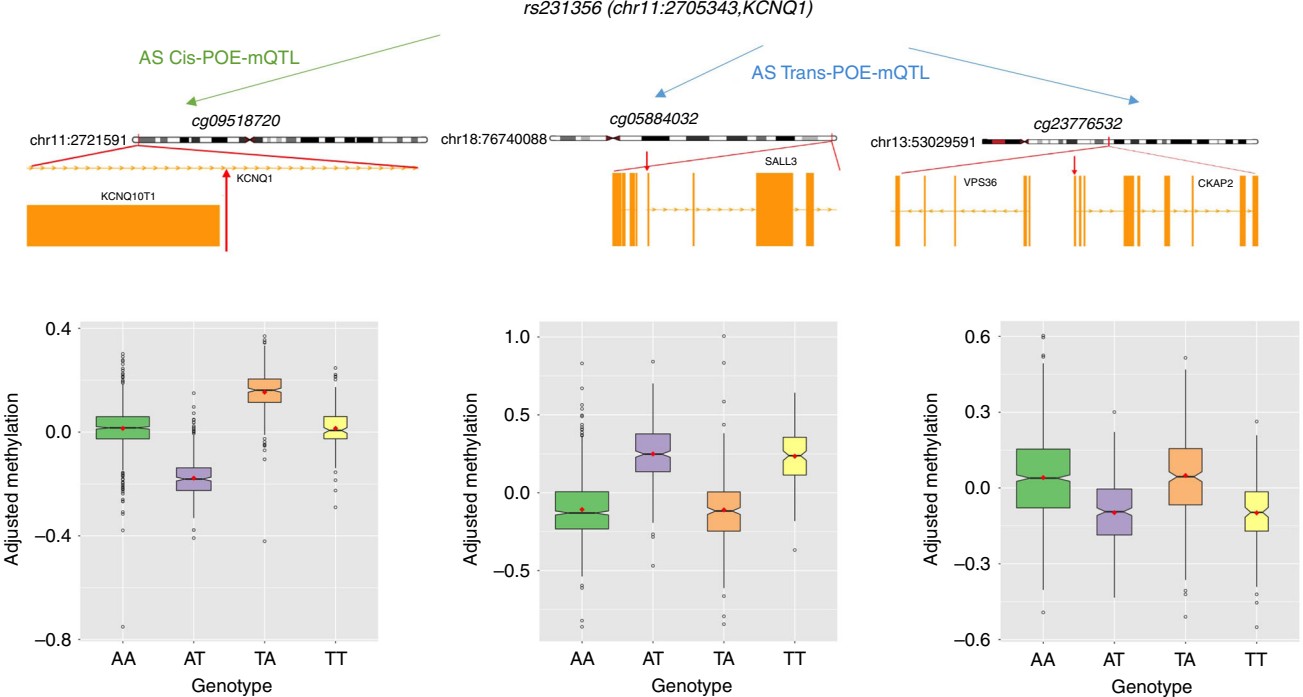

**Fig. 4** Methylation CpGs regulated by SNP rs231356. SNP rs231356 acts both as a *cis*-POE–mQTL and a *trans*-POE–mQTL. Red arrows: location of the CpG in the chromosome. Boxplots show the allelic effects of rs231356 on methylation of three CpG sites (cg09518720, in *cis*, and cg05884032 and cg23776532, in *trans*). Boxplots: centre line, median; box limits, upper and lower quartiles; whiskers, 1.5× interquartile range; points, outliers

**POE–EWAS for associations between complex trait and POE CpGs.** To examine the potential consequences of variation in POE-influenced methylation CpGs on complex traits, we tested their association with 34 traits available in GS:SFHS[24], including anthropometric, cardiometabolic, psychiatric and psychological traits (Supplementary Table 4). Considering the potential biological difference between the CpG group with strong evidence and the group with moderate evidence of POEs (Supplementary Fig. 7), we performed the analyses separately for each group. These analyses identified 22 methylation–trait associations between 14 methylation sites and 10 traits at the multi-trait significance level (Table 4), and 81 methylation–trait associations between 47 methylation sites and 17 traits at the per-trait significance level in at least one group (Supplementary Data 9). These methylation sites were in regions highly enriched in paternal imprinting ($P_{(chi\text{-}squared\ test)} = 2.2 \times 10^{-9}$), particularly for CpGs with strong POE evidence (Supplementary Table 5). Sixteen independent loci (defined as a region containing trait-associated CpGs mapped to the same gene, with between-locus-distance ≥ 1 Mb) were associated with more than one trait (Supplementary Data 9). In both groups, association signals (including those not reaching the significance threshold) for high-density lipoprotein (HDL) cholesterol, vegetable consumption frequency, body mass index (BMI), weight and intelligence (G), were ranked significantly higher than the other traits; waist circumference was ranked significantly higher than other traits only for the strong POE evidence group, whereas creatinine, education, alcohol consumption and blood pressure (systolic) were ranked higher only for the moderate POE evidence group (Mann–Whitney U test; Supplementary Tables 6, 7).

**POE-accounted phenotype-wide association study.** A phenotype-wide association study, accounting for POE, (POE–PheWAS) was performed for 34 phenotypes (Supplementary Table 4) using those POE–mQTL SNPs identified in the previous analyses (for quantile–quantile plot: see Supplementary

Fig. 8). One locus (rs6100212, chromosome 20: 57361064) exceeded phenome-wide significance for an association with waist circumference ($P_{(t\ test)} = 8.57 \times 10^{-7}$, Table 5). The index SNP was located 5′ of the *PIEZO1P2* gene (Fig. 5). The same locus was also associated with waist-to-hip ratio ($P_{(t\ test)} = 1.22 \times 10^{-5}$), BMI ($P_{(t\ test)} = 1.27 \times 10^{-5}$), and body fat ($P_{(t\ test)} = 2.09 \times 10^{-5}$) at the per-trait significance level (Table 5). This *cis*-POE–mQTL was consistent in producing complex imprinting patterns at 12 DNA methylation sites, and also in waist circumference, BMI, body fat and waist-to-hip ratio (Fig. 5). Published GWAS that only accounted for additive genetic effects have failed to detect the association between this locus and the above traits as would be expected for a locus with a complex imprinting pattern (Supplementary Data 10). For the index SNP (rs6100212), we further detected a significant POE-by-sex interaction effect on waist circumference ($P_{(t\ test)} = 1.31 \times 10^{-3}$, Supplementary Table 8). When dividing the sample into age deciles, a nominally significant POE-by-age interaction effect on waist circumference was detected in the 10th decile (age > 47) ($P_{(t\ test)} = 4.91 \times 10^{-2}$, Supplementary Table 8). Combining these results, the largest POE of SNP rs6100212 on waist circumference was detected in females over 47 years old (Supplementary Table 9, Supplementary Fig. 9).

A replication analysis was performed using the subset of UK Biobank (UKB) with inferred parent-of-origin information ($N = 4378$). The significant POE from rs6100212 on waist circumference was not statistically significant in UKB ($P_{(t\ test)} = 0.65$), although a similar trend was observed with the point estimate of POE increasing at older ages and in females (Supplementary Table 10). The lack of significance in UKB data is potentially due to the age difference between the discovery and replication sample (Supplementary Fig. 10), particularly the small number of UKB participants who also have parents in the cohort (which is a pre-requisite for the inference of parental allele origin) and hence have SNP parent-of-origin information categorised as females over 47 ($N = 130$),

**Table 4 Associations between POE CpGs and traits significant at the multi-trait level**

| CpG | Evidence | Chr | Position (bp) | VC | Genic region | Gene name | Trait* | P-value | Est | SE |
|---|---|---|---|---|---|---|---|---|---|---|
| cg11078090 | Strong | 1 | 23878540 | C | Upstream; downstream | E2F2; ID3; LOC101928163 | BMI | $2.22 \times 10^{-7}$ | 0.033 | 0.006 |
| | | | | | | | WC | $5.49 \times 10^{-7}$ | 0.030 | 0.006 |
| cg08259905 | Strong | 3 | 62171428 | P | Intronic | PTPRG | Weight | $1.85 \times 10^{-6}$ | −0.027 | 0.006 |
| cg00329615 | Strong | 3 | 118706648 | C | Intronic | IGSF11 | SBP | $2.48 \times 10^{-7}$ | 0.024 | 0.005 |
| cg10755899 | Strong | 4 | 1772151 | C | Upstream; downstream | FGFR3; TACC3 | HDL | $1.75 \times 10^{-8}$ | −0.029 | 0.005 |
| | | | | | | | BMI | $2.46 \times 10^{-8}$ | 0.020 | 0.004 |
| | | | | | | | %Fat | $4.18 \times 10^{-8}$ | 0.985 | 0.179 |
| cg01290904 | Moderate | 4 | 5708474 | P | Intronic | EVC2 | HDL | $2.36 \times 10^{-7}$ | −0.046 | 0.009 |
| cg11064966 | Strong | 5 | 32506514 | C | Intergenic | None | Weight | $5.11 \times 10^{-7}$ | 0.056 | 0.011 |
| cg12577411 | Strong | 6 | 15551489 | P | Intronic | DTNBP1 | %Fat | $9.60 \times 10^{-9}$ | −2.27 | 0.394 |
| | | | | | | | BMI | $1.03 \times 10^{-8}$ | −0.045 | 0.008 |
| | | | | | | | WC | $3.47 \times 10^{-7}$ | −0.038 | 0.007 |
| | | | | | | | Weight | $3.50 \times 10^{-7}$ | −0.044 | 0.009 |
| cg15773890 | Strong | 6 | 17259549 | P | Upstream | RBM24 | Alcohol | $1.63 \times 10^{-7}$ | −0.315 | 0.060 |
| cg05246100 | Strong | 7 | 55246275 | C | Intronic | EGFR | BMI | $1.02 \times 10^{-6}$ | −0.037 | 0.008 |
| cg11613559 | Strong | 10 | 121577971 | C | Intronic | INPP5F | Alcohol | $2.96 \times 10^{-7}$ | −0.132 | 0.026 |
| cg14391737 | Moderate | 11 | 86513429 | C | Intronic | PRSS23 | Hips | $1.85 \times 10^{-7}$ | 0.014 | 0.003 |
| | | | | | | | Weight | $6.86 \times 10^{-7}$ | 0.025 | 0.005 |
| | | | | | | | BMI | $8.50 \times 10^{-7}$ | 0.023 | 0.005 |
| cg27272202 | Moderate | 12 | 5158794 | P | Downstream | KCNA5 | CREAT | $2.49 \times 10^{-7}$ | 0.049 | 0.009 |
| cg08698721 | Strong | 14 | 101294147 | P | ncRNA intronic | MEG3 | Height | $1.89 \times 10^{-7}$ | −1.08 | 0.206 |
| cg21740139 | Moderate | 17 | 60753158 | P | Exonic | MRC2 | CREAT | $1.83 \times 10^{-6}$ | 0.045 | 0.009 |

The table shows the CpGs displaying POE (CpG), their location (Chr: chromosome and position in bp, location relative to the nearest gene (genic region) and name of the nearest gene(s) (gene name)), the pattern of imprinting detected in the variance component analysis (VC, C: complex, P: paternal), the strength of the evidence supporting the inference of POE (evidence) and the estimated correlation between methylation level and traits (Est, Trait), together with standard errors (SE) and an indication of significance (P-value of t test). *Further details on traits are given in Supplementary Table 4

**Table 5 Significant POE from *cis*-POE–mQTL rs6100212 on phenotypes and CpGs**

| Type | Trait*/CpG | P-value | Est | SE |
|---|---|---|---|---|
| Trait | WC | $8.57 \times 10^{-7}$ | −0.005 | 0.001 |
| Trait | WHR | $1.22 \times 10^{-5}$ | −0.004 | 0.001 |
| Trait | BMI | $1.42 \times 10^{-5}$ | −0.005 | 0.001 |
| Trait | % Fat | $2.09 \times 10^{-5}$ | −0.212 | 0.050 |
| Methylation | cg03837903 | $5.04 \times 10^{-6}$ | 0.028 | 0.006 |
| Methylation | cg04677683 | $2.34 \times 10^{-14}$ | −0.045 | 0.006 |
| Methylation | cg06200857 | $1.96 \times 10^{-4}$ | −0.017 | 0.005 |
| Methylation | cg08091561 | $2.57 \times 10^{-4}$ | −0.020 | 0.005 |
| Methylation | cg09437522 | $5.56 \times 10^{-7}$ | −0.018 | 0.004 |
| Methylation | cg11480267 | $6.06 \times 10^{-6}$ | 0.029 | 0.006 |
| Methylation | cg15160445 | $4.60 \times 10^{-16}$ | −0.047 | 0.006 |
| Methylation | cg23249369 | $1.37 \times 10^{-10}$ | −0.027 | 0.004 |
| Methylation | cg24203465 | $8.53 \times 10^{-10}$ | −0.018 | 0.003 |
| Methylation | cg24617313 | $1.12 \times 10^{-16}$ | −0.159 | 0.019 |
| Methylation | cg25326570 | $2.12 \times 10^{-12}$ | −0.042 | 0.006 |
| Methylation | cg26102503 | $2.51 \times 10^{-24}$ | −0.045 | 0.004 |

*Further details on traits are given in Supplementary Table 4

where the largest POE was detected in GS:SFHS (Supplementary Table 9).

## Discussion

We present here a population-based analysis of POEs caused by imprinting in human DNA methylation. Using a variance component method, we identified 733 independent CpGs (984 total), of which 331 were previously unidentified, where methylation levels displayed an increased full-sibling and/or one-parent–offspring methylation level similarity relative to expectations under additive inheritance patterns, suggesting putative POEs caused by imprinting. For 399 independent CpGs (171 previously unidentified), we identified genetic variants (POE–mQTLs) that regulate the CpGs through POEs. This

provided additional evidence for POEs caused by imprinting on those CpGs (Table 3). CpG sites with putative POEs (candidate POE CpGs) without an associated POE–mQTL displayed distinct enrichment patterns in a range of genomic features and may represent a different biological phenomenon. A large proportion of the identified POE–mQTL associations followed a complex imprinting pattern (Fig. 1). Such a pattern is likely to be undetected in GWAS and classical mQTL studies which only model additive genetic effects, as the complex imprinting pattern produces no phenotypic difference in genotype means associated with allele substitution. We identified 22 significant associations between 14 of these candidate POE CpGs and 10 complex traits. We further examined the parent-of-origin effect of the identified POE–mQTL SNPs on complex traits and identified a locus associated with a complex imprinting effect on waist circumference and related traits that was not detectable by a standard additive effect GWAS. If such complex POEs proved to be widespread, they would contribute towards sibling similarity and hence some traditional pedigree estimates of trait heritability without contributing to SNP-based heritability estimates. Such effects could thus contribute towards the discrepancy between these two heritability estimates, i.e., the missing heritability[30].

The variance component analysis applied to detect POE signatures required data from parents and offspring but has advantages that include: (1) the ability to detect methylation sites with POEs without the need to know where the genetic variants responsible for the effect are located, or even without genotype data (although we used genotypes to construct the genomic relationship matrix, a pedigree-based relationship matrix could be used to replace this); (2) the ability to detect a number of CpGs displaying a complex imprinting pattern, substantially increasing the number of such sites reported with respect to previous studies. In particular, the proportion of sites displaying complex POE is higher among the previously unidentified sites than that for the previously known sites (67% vs. 57%), which means that our method potentially enables discovery of previously unidentified imprinted regions. The reliability of the parent-of-origin

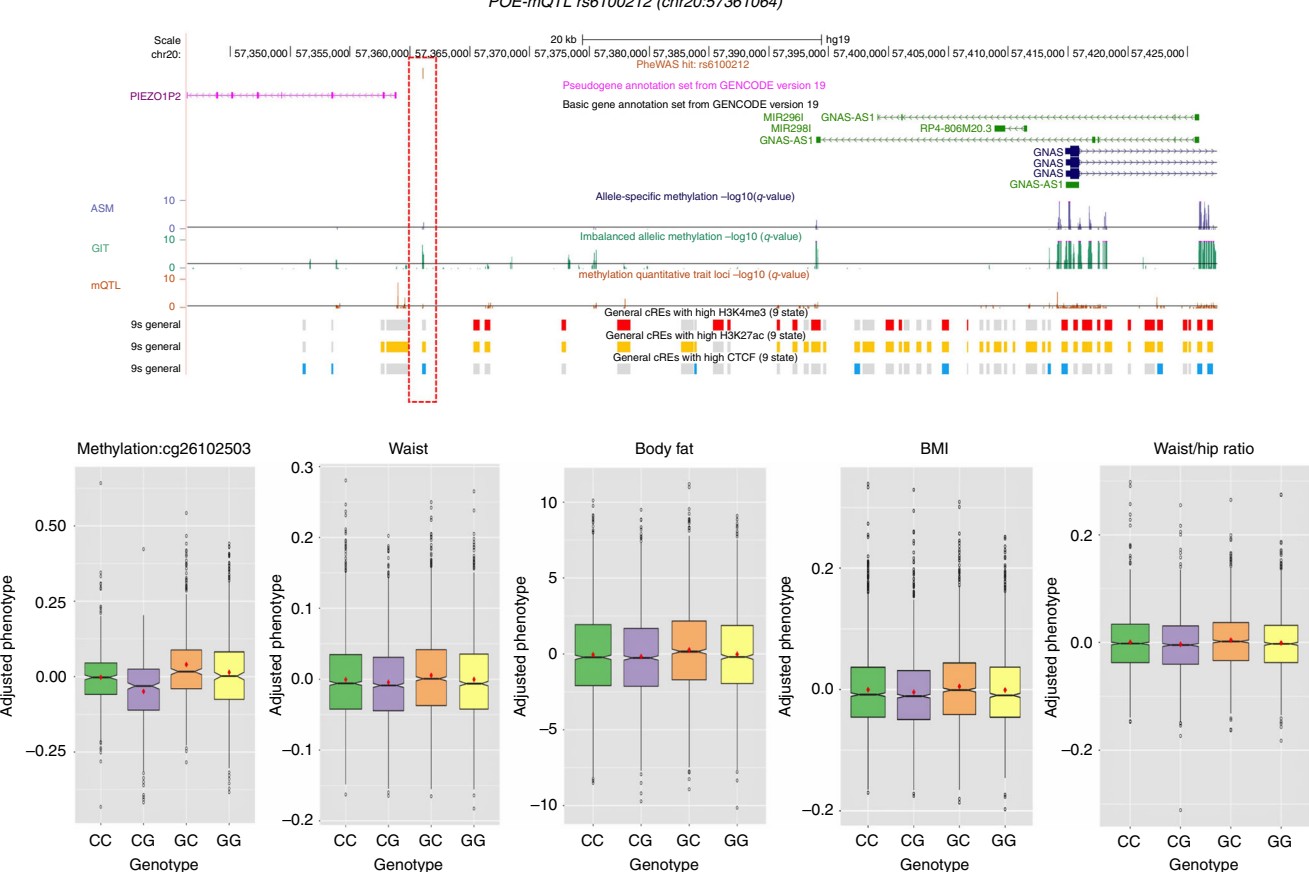

**Fig. 5** CpGs and complex traits regulated by SNP rs6100212. Upper: this SNP was located upstream of gene *PIEZO1P2* and overlapped with H3K27ac and CTCF signals. The SNP was also significant for imbalanced methylation (GIT), but not significant in allelic-specific methylation (ASM) or mQTL (classical additive model) analysis as reported by an independent study. Left bottom: the SNP acted as a *cis*-POE–mQTL for methylation sites causing a complex imprinting pattern (cg26102503 as an example). Middle and right bottom: the SNP was also shown a regulatory role in waist (Phenome-wide significance), BMI, body fat and WHR (per-trait significance), introducing a similar complex imprinting pattern. Boxplots: centre line, median; box limits, upper and lower quartiles; whiskers, 1.5× interquartile range; points, outliers

inheritance patterns inferred for those sites is supported by the fact that of the 331 previously unidentified sites, 51.7% ($N = 171$) have at least one POE–mQTL identified in our study, and the replication rate of 65% for the POE-influenced CpGs identified using a different and genetic-variant dependent method (POE–mQTL) by an independent study[13].

For more than half of the 984 CpGs displaying parent-of-origin inheritance in the variance component analysis, we detected associated genetic variants that influence the methylation variation through a POE. This provided strong evidence that the POEs on those CpGs were caused by imprinting (Table 3). The fact that the CpGs that display global parent-of-origin inheritance were associated with SNPs suggests that (1) the POE–mQTL SNPs were potentially located in imprinted regions and (2) the POEs affecting CpGs detected in this study were potentially the downstream consequence of imprinted states in regions where POE–mQTLs were located (Supplementary Fig. 1). This is supported by the substantial enrichment of detected POE–mQTLs in known imprinted regions and non-genetically regulated imbalanced methylation regions. In addition, a high level of complexity in the association between POE–mQTL SNPs and their regulated CpGs has been revealed in this study. Previous studies suggested that complex imprinting patterns were a consequence of regulation by two genetic regulators with opposite direction of POEs[11], whereas a paternal or maternal imprinting pattern could be regulated by single POE. Here, we observed cases where an individual CpG displayed different imprinting patterns when stratified by genotypes at different independent SNPs (Supplementary Fig. 11), showing that multiple regulations targeted the same CpGs. We also observed cases where single POE–mQTLs introduce different imprinting patterns to different CpG targets (Fig. 4), suggesting that the same SNP interacted with other regulators differently depending on which CpG it targeted.

The enrichment analysis revealed both similarities and differences between the groups of candidate POE CpGs with and without POE–mQTL associations (with strong and moderate evidence of POE). In both groups, the CpGs were in regions enriched in Polycomb repressed regions and depleted in promoters. The group with POE–mQTLs was enriched in noncoding RNA regions. This was in line with previous findings suggesting that Polycomb proteins and imprinted ncRNA acted either cooperatively or independently to regulate imprinted gene clusters[31,32]. The group with POE–mQTL was also located in regions highly enriched in known imprinted region, whereas that was not the case for candidate POE CpGs with moderate evidence, suggesting that the former group mainly reflects POEs that are mechanistically similar with those reported in well-established studies of imprinting, whereas the latter group may reflect a separate group of CpGs influenced by POE, but further work is needed to understand the mechanisms behind our observations. Further differential enrichment between the two groups was observed in genic regions of genes containing published EWAS

hits for various traits. The candidate POE CpGs without associated POE–mQTLs were enriched in genic regions of genes containing smoking-associated methylation CpGs, whereas the candidate POE CpGs with associated POE–mQTLs were in regions enriched in genes previously identified to contain CpGs associated with BMI and alcohol consumption. These results suggested the potential downstream consequence of the variation in these methylation CpGs on these specific traits. This was further supported by the observation that the majority of significantly associated traits for POE-influenced CpGs were metabolic traits (Table 4, Supplementary Data 9). Finally, our POE–PheWAS analysis for POE–mQTL SNPs identified an association between one POE–mQTL and waist circumference and other obesity-related traits. These convergent results imply that a potentially important consequence of identified POE is methylation-mediated variation on metabolic traits.

In contrast to additive genetic models which are used in classical GWAS, the model that we used for POE–mQTL and POE–PheWAS analysis (Table 1) allowed us to detect SNP–phenotype (phenotype being either CpGs or complex traits) associations, in which the SNP causes a phenotypic difference between reciprocal heterozygous groups but not necessarily between the two homozygous groups. Using this model, we identified a locus (tagged by SNP rs6100212) that causes a consistent complex imprinting pattern in both methylation CpGs and waist circumference and related phenotypes. This locus was not significant either in published mQTL analyses, or published GWAS for waist circumference or any other related traits, potentially because of the lack of phenotypic difference between the two homozygous groups (Fig. 5). Our findings are in agreement with those of a recent independent study that found this locus was located in an imbalanced methylation region between paired chromosomes (Fig. 5)[19]. Given the fact that for waist circumference, this POE is particularly strong in the older female group (Supplementary Table 9, Supplementary Fig. 9), replication analysis of this finding in GS:SFHS would be best performed in a sample with a large number of older females with parental origin assigned for the alleles they carry. In UK Biobank, the number of females with parental allelic origin assigned in the >47 class was small (Supplementary Fig. 10), which may explain the lack of significant replication. rs6100212 was located in a regulatory region (supported by high-signal intensity of H3K27ac and CTCF binding as shown in Fig. 5) upstream of the pseudogene PIEZO1P2. Intriguingly, a previous GWAS identified a locus located within an enhancer (GH20H058887, located in the intron of nearby gene GNAS) targeting the same gene (PIEZO1P2) (http://www.genecards.org/) to be associated with waist circumference adjusted for body mass index[33]. Given these convergent lines of research, PIEZO1P2 and its regulatory regions should be treated as targets of obesity-related research.

We combined results from different analyses to evaluate the strength of evidence for POEs caused by imprinting for each CpG (Table 3, Supplementary Data 1). CpGs with strong or very strong evidence of POEs (those with an associated POE–mQTL, N = 586) displayed clear patterns of overall POEs in the variance component analysis and had associated mQTL that drove the observed POE. CpGs in this group should be the focus of future studies targeting the downstream consequences of POE. Whereas candidate CpGs classified as not being supported by strong evidence (those without a POE–mQTL associated, N = 398), should be treated more cautiously. Although they displayed an overall POE pattern and some distinct features compared with the other group (Supplementary Fig. 7), the location and the type of genetic variants causing this pattern are yet to be identified and therefore need further validation. Other factors might also result in the increased full-sibling similarity observed in these CpGs, such as a

full-sibling environmental effect, including some forms of maternal or paternal environmental effects, or a dominance or other non-additive effect. In the POE–mQTL analysis, a SNP dominance effect was also estimated in the model, allowing us to rule this out as responsible for the increased full-sibling similarity for the majority of candidate CpGs. For three CpGs (cg27572120, cg14614539, cg25885219), we failed to detect a POE–mQTL effect, but detected significant dominance effects from at least one SNP. In addition, POEs could be caused by other mechanisms, such as a genetic difference of reciprocal heterozygotes caused by gender-specific biased trinucleotide expansions, or situations where the expression of a locus in the mother (or father) influences the phenotypes in the offspring[11]. The contribution of those mechanisms to the observed POEs should be explored in future analyses.

A limitation of this study is the relative lack of power to detect trans-POE–mQTLs ($N_{individuals}$ = 1668) and to detect SNP–trait associations in our POE–PheWAS ($N_{individuals}$ = 7106), given our sample size. This highlights a challenge for future POE studies because despite the very large size of some cohort studies, a focus on contemporary and unrelated individuals means that very limited parent–offspring data are available. This highlights the need to increase the number and size of family-based cohorts that allow the detection of potentially important sources of variation that may be difficult or impossible to study otherwise. An interesting topic of further research building on our own work would be a systematic investigation of the translation of POE-associated variation in methylation to POE-associated gene expression. Finally, longitudinal or stratified analysis could elucidate the stability of POE patterns at different developmental stages, disease, aging stages and genetic/environmental backgrounds and perhaps most importantly, tissue and cell types.

In conclusion, a methylome-wide scan in 5101 individuals identified 984 candidate CpGs as the targets of POEs caused by imprinting at the DNA methylation level. Of these 984 candidate CpGs, there is strong evidence that 191 are previously unidentified POE-influenced CpGs from 171 independent regions. DNA methylation, genome-wide genotypes and intensive phenotyping data were further combined in a series of comprehensive analyses, where some of the potential causes (POE–mQTLs) and consequences (associated complex traits) of these POEs were uncovered, providing important targets for future studies.

## Methods

**Population samples.** Generation Scotland: The Scottish Family Health Study (GS:SFHS) contains 21,387 subjects ($N_{males}$ = 8,772, $N_{females}$ = 12,615; average age = 47.2 (SD = 15.1)) from ~7000 families who were recruited from the registers of collaborating general practices in Scotland between 2006 and 2011[24]. A subset of 5101 GS:SFHS participants have DNA methylation data (see below). The family structure for that subset includes 1692, 616, 1102 and 306 full siblings, father–offspring, mother–offspring and couple pairs, respectively. The average age of parents is 58 (5%–95%:45–78) and the average age of offspring is 34 (5%–95%:19–53). All components of GS:SFHS received ethical approval from the NHS Tayside Committee on Medical Research Ethics (REC Reference number: 05/S1401/89). GS:SFHS has also been granted Research Tissue Bank status by the Tayside Committee on Medical Research Ethics (REC Reference number: 10/S1402/20), providing generic ethical approval for a wide range of uses within medical research. Participants all gave written consent after having an opportunity to discuss the project and before any data or samples were collected.

UK Biobank data were obtained under application number 19655. We used records on waist circumference on 4378 white-British unrelated individuals for whom parent-of-origin information could be imputed. The UK Biobank project was approved by the National Research Ethics Service Committee North West-Haydock (REC reference: 11/NW/0382). An electronic signed consent was obtained from the participants.

**Genotyping, phasing and imputation in GS:SFHS.** Genotyping data were generated using the Illumina Human OmniExpressExome −8- v1.0 array[34–36]. Phasing was performed using SHAPEIT option–duohmm, and imputation was

performed using the Haplotype Reference Consortium (HRC) reference panel release 1.1[37,38]. A total of 497,401 genotyped common autosomal SNPs and 7,108,491 imputed common SNPs for 19,994 participants passed Quality Control (QC) criteria and were used in the subsequent analyses. Details of QC, phasing and imputation are given in Supplementary Methods. Chromosomal position for markers are based on human genome assembly GRCh37 (hg19).

**DNA methylation data on a subset of GS:SFHS.** DNA methylation data were available for a subset of 5200 participants from the GS:SFHS cohort, as part of the Stratifying Resilience and Depression Longitudinally (STRADL) project[39]. DNA methylation was measured at 866,836 CpGs from whole blood genomic DNA, using the Illumina Infinium MethylationEPIC array. Two formats of data were produced after QC and normalisation: (1) Beta values which measure the proportion of methylation at a given CpG (ranging from 0 to 1); and (2) M values which are the logit transformation of the Beta values. M values were used in downstream analysis as a previous study suggested these to be more statistically robust in analysis[40]. For each methylation site, a linear mixed model was used to pre-correct M values to remove effects of technical factors. The model converged successfully for 639,238 CpG sites, and the resulting residualised-M values were used as DNA methylation phenotypes in downstream analysis ($N_{participants} = 5101$, $N_{CpG} = 639,238$). Details of QC, normalisation, assessment of cell composition, and pre-correction for M-values are given in Supplementary Methods.

**Identification of parent-of-origin of alleles in offspring.** Among GS:SFHS participants with genotype data ($N = 19,994$), there were 2680 trios (i.e., both parents and one offspring), 1185 father–offspring duos, and 3274 mother–offspring duos. We inferred parent-of-origin allelic transmission for 7,108,491 imputed common SNPs (MAF ≥ 0.01) in 7106 of the 7139 offsprings. We compared offspring haplotypes to their parents' using informative loci (i.e., heterozygous) in offspring. We then evaluated the accuracy of the assigned parent-of-origin haplotype at a genotype level, and found an accuracy of over 99.9% across all SNPs. For details, see Supplementary Methods. We used parent-of-origin information in the POE–mQTL analyses and the Phe-WAS described below.

**Assessment of number of independent methylation CpGs.** Methylation levels between CpG sites can be highly correlated. The pairwise correlation in methylation levels between sites was estimated and used to produce a list of independent sites, which we henceforth refer to as index CpGs, using a similar algorithm to that used for LD-clumping (i.e., grouping on the basis of linkage disequilibrium) of SNPs in PLINK, using a window size of 250 kb and a $R^2$ cut-off of 0.1 (Supplementary Methods)[41]. The number of independent index CpG sites and their location was used to compare our results to those described in the literature.

**Variance component analyses of methylation at CpG sites.** A variance component analysis framework based on multiple genomic and family–environmental relationship matrices has been previously developed to dissect phenotypic variation into contributions from additive genetic effect of common SNPs ($h_g^2$), additional additive genetic effect associated with pedigree ($h_k^2$), and a number of shared family environmental effects ($e_f^2$ for nuclear family relationship, $e_s^2$ for full-sibling relationship and $e_c^2$ for couple relationship)[42,43]. We refer to the model applied in these analyses as the GKFSC model. Here, we applied this method to dissect phenotypic variation in DNA methylation levels (measured as residualised M values) for each individual CpG site into these different genetic and family environmental components (Aim 1 in Table 1). We then reparameterised the model to identify candidate methylation CpGs with parent-of-origin inheritance (Aim 2 in Table 1).

**GKFSC variance component analyses.** The GKFSC model[42,43] includes two genomic relationship matrices, **G** (genomic relationship matrix) and **K** (kinship relationship matrix)[42,44], and three environmental relationship matrices, **F** (environmental matrix representing nuclear family-member relationships), **S** (environmental matrix representing full-sibling relationships) and **C** (environmental matrix representing couple relationships)[42,43] (see Supplementary Methods). These five matrices were fitted simultaneously as random effects in a mixed linear model for methylation at each CpG, together with covariates (i.e., age, age², sex, cell-counts for granulocytes, B-lymphocytes, natural killer cells, CD4 + T-lymphocytes and CD8 + T-lymphocytes, season of the visit, appointment time of the day, appointment day of the week) fitted as fixed effects. The model facilitates estimation of the proportion of methylation variation explained by each fitted random effect, while accounting for the effects from the remaining components. The significance of the estimated variance explained by the random effects was tested using a Wald test (one-sided). A Bonferroni correction was applied to account for multiple testing ($N_{test} = 639,238$).

**POE-targeted variance component analyses.** The full-sibling associated variance component ($e_s^2$), modelled in the **S** matrix, may capture not only the shared environmental effect between siblings, but also non-additive genetic effects that increase similarity between siblings. For additive genetic effects, the phenotypic

covariance between parents and their offspring is of similar magnitude to that between siblings, whereas for phenotypes influenced by POEs caused by imprinting, the covariance between parents (one or both, depending on the POE model, see Fig. 1) and their offspring is reduced relative to that between full siblings[45]. Therefore, for CpGs sites for which methylation levels are influenced by POEs caused by imprinting, matrix **S** when fitted simultaneously with the additive genetic components, can capture the additionally increased similarity between full siblings caused by POE (see the Results section), and can be used to identify methylation CpGs potentially influenced by POE (detailed discussion see Supplementary Methods).

There are several possible imprinting inheritance patterns[11], each of which is expected to produce a characteristic covariance structure between parents and offspring and between full siblings (see Fig. 1 for examples). To maximise the power to detect methylation sites influenced by different patterns of POE, we designed three POE relationship matrices that specifically target the POE generated by complex imprinting (**S**, this is the sibling matrix of the GKFSC model), paternal imprinting (**S_P**) and maternal imprinting (**S_M**), respectively (Fig. 1, Supplementary Methods). For each CpG, we compared a model that only includes the additive genetic components **G** and **K** (base model in Table 1), against each of the three alternative imprinting models with one of the POE relationship matrices fitted as random effect, jointly with the genetic additive effects (**G** and **K**). We then selected the alternative imprinting model with the largest significant improvement of model fit, based on a log-likelihood-ratio test (LRT, one-sided, degree of freedom = 1. Supplementary Methods). Multiple testing correction was performed using a false discovery rate (FDR) at 0.05 level ($N_{test} = 639,238$). These analyses were performed in GCTA[46]. The visualisation of results was performed using the R package coMET[47].

**Parent-of-origin effect mQTL analyses.** To locate the loci that cause the POE in the 984 methylation CpGs identified in the variance component analysis, a POE–mQTL analysis was performed by testing 7,108,491 imputed SNPs (MAF ≥ 0.01) against methylation levels for each of these 984 methylation CpGs. There were 1668 offsprings in GS:SFHS with both parent-of-origin assigned to alleles and DNA-methylation data that could be used in our POE–mQTL analysis. For each CpG, methylation values were pre-corrected to account for relatedness by fitting a genomic relationship matrix (**G**) as a random effect and fitting the following variables as fixed effects: age, age², sex, cell count, season of the visit, appointment time of the day, appointment day of the week in a linear mixed model. The residualised M values from the model described above were regressed against three orthogonal genetic effects (two-sided): an additive effect (genotypes coded as 0, 1, 1 and 2 for AA, Aa, aA and aa), a dominance effect (genotypes coded as 0, 1, 1, 0 for AA, Aa, aA and aa), and a POE (genotypes coded as 0, −1, 1 and 0 for AA, Aa, aA and aa) (Table 1)[13]. SNPs showing a significant POE for a methylation CpG and within less than 1 Mb from that CpG were defined as cis-POE–mQTLs[8]. SNPs showing a significant POE for a methylation CpG located more than 5 Mb away from that CpG were defined as trans-POE–mQTLs[48]. SNPs located between 1 and 5 Mb from their POE-associated methylation CpG were not considered. To determine the significance threshold for association, a permutation-based multiple testing correction was performed for cis-POE–mQTLs and trans-POE–mQTLs analyses separately at the FDR ≤ 0.05 level. For the permutation test, individual identifiers were shuffled and the correlation structure between SNPs and between CpGs was retained[9,48]. Ten replicates were used to establish a stable distribution of the test statistic under the null hypothesis, as suggested in previous studies[9,48], which led to an estimate of the FDR ≤ 0.05 p-value threshold of $3.6 \times 10^{-4}$ for cis-POE–mQTLs and $2.19 \times 10^{-9}$ for trans-POE–mQTLs. PLINK was used to produce a set of independent POE–mQTLs by clumping POE-associated SNPs within a window size of 250 kb around the most significant associated SNP (the index SNP) with an $R^2$ threshold of 0.1 and a p-value threshold of 1 for $P_{POE–mQTL}$[41].

**POE–EWAS for associations between complex traits and POE CpGs.** To assess the phenotypic effect of variation in methylation levels at the 984 CpGs identified as potentially influenced by POE, methylation levels at these sites were correlated with phenotypic values for 34 anthropometric, cardiometabolic, psychiatric and psychological traits available in GS:SFHS (details of traits and pre-processing are in Supplementary Table 4)[24]. A linear mixed model was used to pre-correct each of the 34 traits for covariates (age, age², sex, clinic) by including them in the model as fixed effects, and for relatedness by fitting the **G** and **K** matrices as random effects, following previous work[49]. Methylation levels were pre-adjusted for cell count, season of the visit, appointment time of the day, appointment day of the week, never/ever smoking and pack years of smoking. Pairwise association tests were performed by regressing each pre-corrected phenotype against adjusted methylation levels at each CpG site using a linear regression model (tested two-sided). A principal component analysis of the 34 traits revealed that the top 27 principal components explained more than 95% of the variation and any component beyond it has an eigenvalue <0.5 (Kaiser's rule), hence the number of independently tested traits ($N_{inde\_traits}$) was estimated to be 27. Bonferroni-based multiple testing correction was performed, with the p-value significance threshold for multiple traits level estimated to be $1.88 \times 10^{-6}$ ($N_{test} = 27*984 = 26,568$), and for per-trait level estimated to be $5.08 \times 10^{-5}$ ($N_{test} = 984$).

**POE-accounted phenotype-wide association study**. To explore if the POE–mQTL were also associated with POE effects on phenotypes, the 7106 GS: SFHS offspring with parent-of-origin assigned alleles were used in a POE–PheWAS for 34 traits (trait list and pre-correction process were the same as used in the previous section; see Table s1). Only SNPs that were significant in the POE–mQTL analyses described above were used (N$_{total\_snps}$ = 38,122, N$_{inde\_snps}$ = 1895) in this analysis. The same regression model applied to the residualised M values in POE–mQTL analysis was used to perform the POE–PheWAS on the 34 pre-corrected phenotypes (Table 1). As above, the POE–PheWAS model accounts for additive effects, dominance and POEs (tested two-sided). The number of independently tested traits (N$_{inde\_traits}$) was estimated to be 27 (see the previous section), and multiple test correction was performed using Bonferroni method (N$_{test}$ = N$_{inde\_snps}$*N$_{inde\_traits}$ = 51,165).

In order to validate the PheWAS results obtained in GS:SFHS, UK Biobank (UKB) data (N$_{participants}$ = 501,726) were used in a replication analysis[50]. For details of sample information, the identification of nuclear family members, phasing, imputation and QC and parent-of-origin information assignment, see Supplementary Methods. Parent-of-origin information was assigned to alleles of the target SNP (i.e., significant in the GS:SFHS PheWAS) for 4378 white-British unrelated UKB offspring (Kinship coefficient ≤ 0.05) used in the replication analysis (Supplementary Methods). Log-transformed waist circumference was tested for POE using a linear regression model accounting for additive effects, dominance and POEs as well as a number of covariates (age, sex, processing batch, assessment centre, genotype array and top 15 principal components of ancestry) as fixed effects (tested two-sided).

**Functional enrichment of POE-influenced CpGs**. To further characterise the methylation CpGs identified as displaying POEs, ANNOVAR[51] was used to annotate CpGs to regions of (1) different chromatin- and histone-modification states, as DNA methylation dynamics is associated with altered chromatin structure[52], and coupled with histone modifications in relevant tissues[53], and transcription factor binding sites. A lymphoblastoid cell line (GM12878) and an immortalised myelogenous leukaemia cell line (K562) were used in this annotation as they are the two cells produced from blood among primary cell lines with abundant annotation information in the ENCODE project[54]. Methylation CpGs were also annotated to (2) regions that are significant in published GWAS and EWAS and (3) substructure regions of genes (for databases used see Supplementary Methods). Fisher's exact test was used to test for enrichment/depletion of POE-influenced CpGs sites in target annotations. The Bonferroni method was used for multiple testing correction (N$_{test}$ = 212 (see Supplementary Data 2), p-value threshold of significance = $2.36 \times 10^{-4}$).

**Gene set-based enrichment of POE-influenced CpGs**. A further characterisation of the POE-influenced methylation CpGs involved annotating these to genes and then testing for enrichment of those annotated genes in specific gene set. ANNOVAR[51] was used to annotate CpGs to genes. A CpG site was assigned to its nearest gene if it was located between 5 kb of the gene's transcription start site (TSS) and 1 kb distance from the transcription end site (TES). The online tool DAVID was used to perform an enrichment analysis in GO-ontology terms, biological pathways, GAD (Genetic Association Database) diseases, protein domains and interactions[55]. The enrichment test was performed using the EASE score test (a modified Fisher exact test which is more conservative than the standard Fisher exact test) to see whether the proportion of genes falling into the tested annotation differs in a target group compared with the background group.

## Data availability

We have made summary statistics of significant results in all tests available in Tables and Supplementary Information. Full summary statistics are available from authors upon reasonable request and consistent with participant consent. Data are available from the MRC IGMM Institutional Data Access/Ethics Committee for researchers who meet the criteria for access to confidential data. GS:SFHS data are available to researchers on application to the Generation Scotland Access Committee (access@generationscotland. org). The managed access process ensures that approval is granted only to research which comes under the terms of participant consent which does not allow making participant information publicly available. UK biobank data are available from UK Biobank, for details see: http://www.ukbiobank.ac.uk/wp-content/uploads/2012/09/Access-Procedures-2011-1.pdf.

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

## Acknowledgements

Generation Scotland received core support from the Chief Scientist Office of the Scottish Government Health Directorates [CZD/16/6] and the Scottish Funding Council [HR03006]. Genotyping of the GS:SFHS samples was carried out by the Genetics Core Laboratory at the Wellcome Trust Clinical Research Facility, Edinburgh, Scotland and was funded by the Medical Research Council UK and the Wellcome Trust (Wellcome Trust Strategic Award STratifying Resilience and Depression Longitudinally (STRADL) Reference 104036/Z/14/Z). The research was conducted using the UK Biobank Resource under Application number 19655. We acknowledge the Medical Research Council (MRC) UK for funding (grants MC_PC_U127592696 and MC_PC_U127561128). AB gratefully acknowledges funding support from a Wellcome Trust PhD training fellowship for Clinicians, The Edinburgh Clinical Academic Track (ECAT) programme (204979/Z/16/Z). We are grateful to all the families who took part, the general practitioners and the Scottish School of Primary Care for their help in recruiting them, and the whole Generation Scotland team, which includes interviewers, computer and laboratory technicians, clerical workers, research scientists, volunteers, managers, receptionists, healthcare assistants and nurses.

## Author contributions

C.S.H. and Y.Z. conceived and designed the study presented in this paper. A.M.M., C.S.H., I.J.D., D.P. and K.L.E contributed to conceive and design the study population and data recording. Y.Z. conducted the analyses. C.S.H., Y.Z., P.N., C.A., C.X., T.S.B., D.W.C., A.B., D.S., A.T., O.C., K.R. and R.M. contributed the statistical analysis methods. R.M.W., K.E., S.W.M., C.H., A.C., T.S.B., D.W.C., J.F.W., C.A. and R.N. managed and maintained the data and performed the quality control. Y.Z., C.S.H., C.A. and P.N. wrote the paper. All authors discussed results, read and approved the final paper.

## Additional information

**Competing interests:** The authors declare no competing interests.

