## [Peer Review File · Nature Communications]

Reviewer #1 (Remarks to the Author):

Zeng et al conduct a genome-wide scan for Parent-of-Origin Effects in DNA methylation using a large Scottish cohort. They identify a large number of potential POE using a novel variance component methods and follow these up by localizing the genetic variants underlying these effects.

My main concern is the use of a FDR in determining the number of significant effects. FDRs are known to have much high false positive rates in the presence of correlated data, such as DNA methylation. For example, there were 78 sites with a significant full sibling relationship at a Bonferroni significant level was 78. However, when testing for complex imprinting type POE (the same model) using FDR, there are 606 sites where that model is the most significant. In fact, the least significant POE effect from the VC model is 9.1×10^{-5} (Table S3). Taking that as a 5% FDR threshold is equivalent to taking a Bonferroni correction for 549 sites and is clearly not stringent enough.

The QQ plot for the PheWAS shows the most significant SNP being less significant than expected. Does this SNP survive a Bonferroni correction for the full number of traits? Determining 27 independent traits out of 34 by looking at the number of PCs that explain 90% of the variation is not a valid approach. If the 34 traits were completely independent, you could not correct 31 traits because that captures 90% of the variation.

Minor comments:

Describe the relationships in the GS:SFHS in more detail. i.e. provide the number of pairs of M-O/F-O/SP within the dataset. What was the age distribution of parents and offspring?

Were cell counts used to correct methylation estimated or measured?

Why were different covariates used to correct methylation of the testing of association between POE DNAm and complex traits?

When testing for POE mQTL, why was methylation corrected only for G (GRM) and not K (kinship matrix)?

Why is F (nuclear-family environment) not included in the POE test model? Similarly C (couple effect)?

What is the correlation between DNAm and WHR for the sites with POE through rs6100212?

The claim that POE with WHR is strongest with females aged > 47 is suggested as the reason for lack of replication in UKBiobank. However, from Table s14 the effect size in that category is not significantly different to any other.

Other possible mechanism of similarity among family members other than POE are largely dismissed. There are some other plausible mechanisms for significance of the "complex POE" effects - e.g. it is likely that pairs of siblings have experienced a more uniform environment during development than they have with their parents. Improve discussion of this limitation is warranted.

The figures have relatively small text for their internal labels, and often labeling could be improved (e.g. Fig 5 "nuclear_family_pairwise_cg...")

Reviewer #2 (Remarks to the Author):

Zeng et al.

'Parent of origin genetic effects on methylation in humans are unexpectedly common and influence complex trait variation.'

Zeng et al. use a unique dataset to identify SNPs whose parent of origin is associated with nearby methylation variation and further associate this with phenotypes. They combine classic quantitative genetics variant components analyses with GWAS to model POE. The fundamental assumption of their method is that covariance between P-O < S-S means a parent of origin effect.

You assume imprinting throughout the manuscript without suggesting other mechanisms besides imprinting (maternal effects, paternal effects) that cause POE. Imprinting is an allele-specific phenomenon and you do not test for allele specific methylation. POE does not always mean imprinting. Further, differential methylation by nearby reciprocal genotype does not mean imprinting. Please clarify or justify your use of imprinting (which implies mechanisms that are alluded to but not tested)

You overlap your methylated regions with another study (ref 18) that tested for ASM and you use the overlap to validate you are identifying imprinting. However – this other study looked at multiple cell types (including adipose and skeletal muscle) did you verify that the overlaps you find are using

the same cell types? What percentage of their bisulfate converted bases needed to overlap your methylated regions to call it an overlap?

The assumption that differential methylation associated with a SNP is a consequence of POE rather than a driver does not seem valid and is not tested. Please clarify the logic.

What does whole blood have to do with the phenotypes you are mapping? You are assuming whole blood methylation status is a good proxy for POE on the phenotypes you test. Please justify.

You corrected for cell counts of some different cell types (immune cells) but there are many different cell types in whole blood and different cell types are known to have differing methylation profiles. Did you test the alpha diversity of the samples? Perhaps the genotypic diversity is actually driving cell type variation (and not methylation) and this is what you are detecting in most of your associations? Parent of origin effects are likely only a subset of the effects that you detect ... and likely a much smaller subset than claimed in the manuscript.

You do a lot of overlapping of your regions with regions reported in other data sets, but you do not test whether the SNPs or mQTL you identify fall in known imprinted control regions or CpG islands. The argument that you are indeed detecting imprinted POE would be much more valid if you homed in on some resolved imprinting control regions (for example PEG3 or IGF2) as proof of principal. Then test to determine if you see the same patterns as you see in the novel sites you detect.

Please justify the different methods of multiple tests correction you do at different stages of analysis.

Why did you perform a Ward test and not a more standard LRT in your initial mixed model?

You tried to replicate a very complex context-dependent finding (women >47) and it didn't replicate. This isn't a surprise as you need to subset the data very specifically – do you have any general, non-context dependent loci you could try to replicate? A general effect is not going to have the sample size issues you suggest underlie your lack of replication.

The association of BMI with older women (who are likely peri- or post-menopausal and will have a greater BMI) is not a surprise given that the >47 year old cohort would be enriched for peri- and postmenopausal women. You should perform this analysis correcting for peri- or post-menopausal state to determine if your association holds, if it is truly with BMI, or if it is an artifact of another physiological process.

How many context dependent associations did you find and how many were 'general' associations across the entire population tested?

The text should be edited for brevity. A lot of the language in the manuscript is exactly the same as that in the supplement and a lot of modifying words are unnecessary and should be completely eliminated.

"Given the fact that there are multiple possible imprinting inheritance patterns [34], each of which is expected to display a different covariance structure between parents and offspring and between full-siblings (Figure 1)." What does this sentence mean?

Figure 1 legend should say "expected" covariance structures rather than "corresponding".

Reviewer #3 (Remarks to the Author):

The authors use a well defined and well studied Scottish cohort where phenotypes and DNA methylation data have been collected. Specifically there are 1,668 offspring in the cohort that have both parent-of-origin assigned to alleles and DNA-methylation data that were used in the POE-mQTL analysis. The authors modelled parent of origin effects using phenotypes, variants and EPIC data with a model choice that offered maternal, paternal or complex classification of effects.

The authors define imprinting and P-O-O effects and note that there are only a few imprinted gene in the genome but that parental affects can be the result of regulatory variants around the genome and go on to show this in the data analysis.

The analysis looked for CpGs potentially influenced by POEs, then complex traits associated with DNA methylation levels.

The parameters of the study subjects (collection day and so forth), appear multiple times, this could be parred back a bit to reduce repetition.

The authors talk about assigning CpGs to regions of "different chromatin and histone modification states", this will require a bit more explanation and rationale, other than you can do it because there

are data in the public domain. Consideration for matched tissue is not mentioned, why this is useful is not discussed.

Not surprisingly, the study found genetics as the largest factor in methylation variation. The “shared environmental effects” measured were to do with living in the same place as family members of partners.

Does a genetic variant ‘regulate a CpG in a parent of origin fashion’? Could the authors use a more mechanistic way of terming this relationship?

“The identified CpGs influenced by POEs were enriched in polycomb repressed regions and ncRNA”. The CpGs would be in REGIONS enriched in polycomb, rather than the CpG site itself.

“These CpGs also enriched in genes”?? doesn’t make sense.

“PIEZO1P2 and its regulatory regions should be treated as targets of obesity-related

Research” or maybe targets for post-menopausal (eg hormonal) differences compared to pre-47 year old females?

Overall the study makes use of a strongly curated cohort and applies an epigenome style analysis to the genetic and phenotypic data to model parent of origin effects and identify regions of interest. The study places the effects into categories and looks to explain effects in a more consistent way than some others have done (which it cites). The study also highlights a waist size phenotype which is one of the main findings of the paper.

We thank the reviewers for their comments and apologise that our previous version was unclear in places. Below we show reviewers' comments (**in bold**) and our responses (in plain text).

Reviewer #1 (Remarks to the Author):

Q: Zeng et al conduct a genome-wide scan for Parent-of-Origin Effects in DNA methylation using a large Scottish cohort. They identify a large number of potential POE using a novel variance component methods and follow these up by localizing the genetic variants underlying these effects.

My main concern is the use of a FDR in determining the number of significant effects. FDRs are known to have much high false positive rates in the presence of correlated data, such as DNA methylation. For example, there were 78 sites with a significant full sibling relationship at a Bonferroni significant level was 78. However, when testing for complex imprinting type POE (the same model) using FDR, there are 606 sites where that model is the most significant. In fact, the least significant POE effect from the VC model is 9.1×10^{-5} (Table S3). Taking that as a 5% FDR threshold is equivalent to taking a Bonferroni correction for 549 sites and is clearly not stringent enough.

A: The EPIC array DNA methylation data does have correlated probes [1], however, a consequence of this is that the number of independent tests is lower than the number of tests performed, which would imply an over-correction if using Bonferroni correction. When performing genome-wide analysis, FDR is the most widely used method of multiple testing correction for gene expression analysis [2], and it is also used by a number of studies for DNA methylation [3-5]. We disagree with the reviewer's comment that: "FDRs are known to have much high false positive rates in the presence of correlated data, such as DNA methylation." Substantive literature suggests that FDR is robust to positive dependencies between tests as are found in most genomic (and many other) studies (e.g. Benjamini and Yekutieli, *Annals of Statistics* 2001, 29, 1165–1188, subsequent references to this study and Wikipedia). In our study, we are doubly protected as we have a two-stage analysis for the identification of POE on methylation, the variance component analysis as the first stage, used for candidate site identification for follow-up in the POE-mQTL analysis as the 2nd stage, which provides additional evidence for the candidates identified in stage 1. FDR is used in our variance component analysis and in the POE-mQTL (following published mQTL studies [6, 7]). In the latter we used a permutation based FDR, which accounts for the correlation structure of the data [7]. In these two situations Bonferroni would have been over conservative. We checked the overlap between our POE-mQTL results with in a recently published independent study (Cuellar Partida, G., et al., *Genome-wide survey of parent-of-origin effects on DNA methylation identifies candidate imprinted loci in humans*. *Hum Mol Genet*, 2018) and we observed a replication rate of 65% (N_cpg=129) (this result is newly added in this version of the paper in lines 546-558). The FDR is designed to control the proportion of false positive results amongst those declared significant and hence along with the true positives we expect a small proportion of false positive results at the first analysis stage, but we use this first stage to reduce the number of probes (hence number of analyses and tests) that we take for the second stage of the analyses, as the power of POE-mQTL analysis, particularly trans-POE-mQTL analysis, is limited. It is also possible that the POE on

those CpGs that are significant at the first stage of the analysis but not the second are introduced through different path beyond that accounted for by the POE-mQTL model, as we indeed observe some characteristic differences in candidate CpGs from this group (Figure 6).

We tried to clarify all these points in the paper:

- 1) We adjusted the structure of the paper by moving the POE-mQTL section forward (lines 481-575);
- 2) We clarified the two-stage analysis in Table 1, highlighting the different strength of evidence for POE in Table 3;
- 3) We added enrichment and association analyses separately for the candidate CpG group (from the first stage of the analyses) with POE-mQTL detected and the candidate CpG group without POE-mQTL detected to characterise the two groups (lines 560-575, 582-598, figure 6) and
- 4) We highlighted in the discussion section that the candidate CpGs without POE-mQTL detected should be treated with caution and need future validation (lines 778-783).

Q: The QQ plot for the PheWAS shows the most significant SNP being less significant than expected. Does this SNP survive a Bonferroni correction for the full number of traits? Determining 27 independent traits out of 34 by looking at the number of PCs that explain 90% of the variation is not a valid approach. If the 34 traits were completely independent, you could not correct 31 traits because that captures 90% of the variation.

A: The deflation of test statistics as shown in the QQ plot suggested that the test is over-conservative, not inflated. The phenotypes used in the PheWAS are correlated, so the number of independent tests is smaller than the number of test performed, and that should be accounted for when adjusting for multiple testing. Bonferroni is an already very stringent method and not accounting for this can introduce over-correction. In fact, a number of published PheWAS studies used FDR [8-10], which could be justified but is a less stringent correction method than our suggested PCA adjustment. We added note in the paper that the top 27 principal components explained more than 95% of the variation and any component beyond it has an eigenvalue < 0.5 (Kaiser's rule) (lines 337-338).

Minor comments:

Q: Describe the relationships in the GS:SFHS in more detail. i.e. provide the number of pairs of M-O/F-O/SP within the dataset. What was the age distribution of parents and offspring?

A: In GS:SFHS, the subset with DNA methylation data (N=5101) includes 1692, 616, 1102 and 306 full sibling, father-offspring, mother offspring and couples, respectively. The average age of parents is 58 (5%-95%: 45-78) and the average age of offspring is 34 (5%-95%: 19-53). We have added this information in the methods section of the new version of the paper (lines 156-159).

Q: Were cell counts used to correct methylation estimated or measured?

A: They were estimated using the estimate CellCounts() function in R package minfi[8] (see supplementary text, lines 42-44).

Q: Why were different covariates used to correct methylation of the testing of association between POE DNAm and complex traits?

A: We used two main sets of covariates for correcting for factors influencing CpG variation (Cov-set1) and trait variation (Cov-set2):

Cov-set1: age, age², sex, cell-counts for granulocytes, B-lymphocytes, natural killer cells, CD4+ T-lymphocytes and CD8+ T-lymphocytes, season of the visit, appointment time of the day, appointment day of the week

Cov-set2: age, age², sex, clinic

In Cov-set1 we are adjusting methylation levels for known factors that impact CpG variation to reduce residual noise. For some of the analyses the methylation values were pre-corrected for those covariates and the residuals were used as trait values in the appropriate models (e.g., POE-mQTL analysis, VC analyses).

Cov-set2 was used to adjust traits used in the trait vs POE CpG and PheWAS for known factors that impact trait variation to reduce residual noise.

We also pre-corrected methylation and the traits for genetic random effects by including in the models one or two genetic matrices (G and K). See answer of the next question below.

Q: When testing for POE mQTL, why was methylation corrected only for G (GRM) and not K (kinship matrix)?

A: In POE-mQTL analysis, the sample size is small (N=1.6K), which means fitting two matrices (G+K) in REML can lead to convergence problems. Whereas in the trait vs POE CpG and PheWAS analyses, with a sample size of N=7K, fitting the two matrices becomes feasible. G and K are highly correlated and it is only possible to discriminate between them with a reasonable sample size, thus the impact of including G versus including G+K is generally very limited and previous research suggests that including G is equivalent to including G+K in most cases (cite source), as G absorbs variation otherwise explained by K if K is not fitted.

Q: Why is F (nuclear-family environment) not included in the POE test model? Similarly C (couple effect)?

A: Fitting more matrices in the first round of analyses means much higher computational burden. Additionally, the available sample size (N=5K) means fitting a larger number of matrices will increase convergence problems. The F effect has been tested in the full model

for all measured CpGs including those identified POE influenced CpG candidates (supplementary table s19), since F did not capture significant variance for any sites, we are confident on removing it for subsequent analyses. The same is true for the C matrix which additionally has very low co-linearity with the S matrix, so not including it in our analytical model should not affect the identification of POE or increase the number of false positives.

Q: What is the correlation between DNAm and WHR for the sites with POE through rs6100212?

A: We have tested the correlation between those DNAm and WHR, and none of them reached significance after multiple-testing correction (for either FDR or Bonferroni corrections).

Q: The claim that POE with WHR is strongest with females aged > 47 is suggested as the reason for lack of replication in UKBiobank. However, from Table s14 the effect size in that category is not significantly different to any other.

A: The effect size in that category is the highest in both GS:SFHS and UK Biobank (although in UK Biobank it is not statistically significant). This is now included in table s17 for GS:SFHS and s18 for UK Biobank.

Q: Other possible mechanism of similarity among family members other than POE are largely dismissed. There are some other plausible mechanisms for significance of the "complex POE" effects - e.g. it is likely that pairs of siblings have experienced a more uniform environment during development than they have with their parents. Improve discussion of this limitation is warranted.

A: We agree that sibling similarity can be increased for a variety of reasons and we accounted for this in our analyses. Thus, when looking at all the analyses presented in the paper we are quite confident that there is compelling evidence pointing that POE are the best explanation for most of our results.

For the 378 candidate POE CpGs where the best VC model was GKSp (paternal) or GKSm (maternal) (Table 3), models including component increasing between-Siblings+paternal-offspring or between-Siblings+paternal-offspring similarity explained the data better than when the model included just the Sibling component (where inclusion of S accounts for exactly the effect of a uniform sibling environment to which the reviewer refers). For those CpGs, POE (or paternal/maternal effect) is a more plausible model than an environmental effect shared between siblings. For 214 of the candidate POE CpGs where the best VC model was GKS we also detected POE-Mqtl effects, or the 9 loci were very close (within 2kb) to known imprinted regions, suggesting that POE is more likely than shared sibling environment. Finally, the candidate CpGs where the best VC model was GKS with no additional supporting evidence for a parent-of-origin effect (264 CpG sites) were considered less reliable and subjected to future investigation and validation but we believe still merit being reported with this caveat.

We have now further clarified this in the discussion, including other possible mechanisms that may contribute to the ‘complex’ pattern. The modified version of the paper now includes:

- 1) Analyses aiming to characterise the two categories of CpG candidates (with POE-mQTL identified *versus* candidates without POE-mQTL identified) (lines 560-575, 582-598);
- 2) Sections highlighting both in the results (Table 3) and discussion section that the candidate CpGs without POE-mQTL detected should be considered as needing future confirmation of any PoE (lines 778-783);
- 3) Discussion on other mechanisms that may introduce the observed POE-like patterns (lines 783-794).

Q: The figures have relatively small text for their internal labels, and often labeling could be improved (e.g. Fig 5 "nuclear_family_pairwise_cg...")

A: Thank you for the suggestion. We have now modified the labels and text in the figures so they are more readable.

Reviewer #2 (Remarks to the Author):

Zeng et al.

‘Parent of origin genetic effects on methylation in humans are unexpectedly common and influence complex trait variation.’

Zeng et al. use a unique dataset to identify SNPs whose parent of origin is associated with nearby methylation variation and further associate this with phenotypes. They combine classic quantitative genetics variant components analyses with GWAS to model POE. The fundamental assumption of their method is that covariance between P-O < S-S means a parent of origin effect.

Q: You assume imprinting throughout the manuscript without suggesting other mechanisms besides imprinting (maternal effects, paternal effects) that cause POE. Imprinting is an allele-specific phenomenon and you do not test for allele specific methylation. POE does not always mean imprinting. Further, differential methylation by nearby reciprocal genotype does not mean imprinting. Please clarify or justify your use of imprinting (which implies mechanisms that are alluded to but not tested)

A: Firstly, the reviewer suggests that “The fundamental assumption of their method is that covariance between P-O < S-S means a parent of origin effect.” The parent-offspring correlation being less than the sibling correlation is one potential symptom for a POE and we agree that for this specific inequality other explanations are possible thus in the reported work further analyses were performed to explore whether this is consistent with POE or with an enhanced common sibling environment effect.

To further answer your questions,

- 1) Imprinting is a phenomenon where a specific genomic region is selectively silenced according to the sex of the parent from which it was inherited (so the expression of a gene is paternal or maternal allele specific) [11]. One of its consequences, is to mask the specific genetic effect of an allele inherited from one of the parents so the offspring's phenotype depends upon the specific effect of the (non-masked) allele inherited from the other parent. This 'masking' of one of the inherited parental alleles could impact the methylation level of some CpGs, and this is the target of our study. Thus our study depends upon the combination of two factors: a) genetic variation that influences the level of methylation at specific CpG sites in the genome, and b) the expression of these alleles being influenced (generally masked or otherwise) based on the sex of the parent from which they are inherited. Both of these factors must be present for us to detect parent-of-origin effects on methylation at specific CpG sites. For example (Figure 1), for a given genetic variant with two possible alleles A/C located in an imprinted region, with one of the alleles (A) increasing the methylation level of a target CpG: the allelic effect of allele A on methylation level is influenced both by the dosage of A alleles and the parent of origin of A alleles. Such imprinting-introduced 'masked' genetic effects have been tested and validated in a number of published studies and reviews [12-14]. In order to clarify this in the paper we added a more detailed explanation of the terms we are referring to in the new version of the paper (lines 72-74, 77-79, 80-82) and a new figure (Figure 1) .

- 2) The first stage in the analysis is identification of CpGs potentially influenced by the genetic effect from variants masked by imprinting (which we call POE). We can detect those CpGs using variance component (VC) analysis because these effects generate differential correlations of methylation levels of target CpGs between pairs of individuals with same level of genetic similarity (full-siblings *cf* father-offspring *cf* mother-offspring). The pattern of similarity is different to that expected for an additive genetic effect, and can be identified using the VC approach modelling POEs. This analysis aims to identify the variation introduced by different patterns of similarity at a genome-wide level (detecting CpGs influenced by POE-masked genetic effects without prior information on the location of the mediating genetic variants). We agree that other effects (such as family environment, maternal environment or non-additive genetic effects other than POE) can produce the same patterns of similarity between family members, hence we subsequently applied a POE-mQTL model to localize those variants. This analysis successfully identified SNPs that mediate POE in 586 out of 984 identified CpG-POE-affected candidates. This demonstrates that VC analysis to identify POE on those CpGs is a useful method to select potential candidates (Table 1). Using VC analysis, we have now also replicated POE-introduced CpGs identified solely through POE-mQTL analysis performed in an independent study (ALSPAC) in GS:SFHS (by variance component analysis) with a replication rate of 65% (N_cpg=129), which strongly supported the validity of the VC method in identification of POE. These new results have been included now included in the paper (lines 546-558).

We have further clarified the two stage analysis in the paper. CpGs for which POE have been significantly identified in both stages can be confidently considered to be

influenced by POE, whereas CpGs which are only significant in variance component analysis should be treated with more caution and need future validation as they could result from effects other than POE. However, we do not want to simply ignore candidate CpGs that did not attain significance in the POE-mQTL analysis, as the power of POE-mQTL analysis and particularly of trans-POE-mQTL analysis is limited. Furthermore, it is possible that the POEs on those CpGs are introduced through a different path to that accounted for by the POE-mQTL model as we observed some characteristic differences in candidate CpGs from this group (line 560-575, 592-598, Figure 6 in the new version of the paper). Therefore, 1) we have adjusted the structure of the paper by moving the POE-mQTL section forward (to directly follow the variance component analysis (lines 481-575), 2) we have clarified the two-stage analysis in Table 1, highlighting the different strengths of evidence for POE in Table 3, 3) we have added enrichment and association analyses separately for the candidate CpG group with POE-mQTL detected and the candidate CpG group without POE-mQTL detected to characterise the two groups in lines 560-575, 582-598, figure 6) we have highlighted in the discussion section that the candidate CpGs that without POE-mQTL detected should be treated with caution and need future validation in lines 778-783.

- 3) We acknowledge that POE may not always result from imprinting and there might be other mechanisms that also introduce the observed pattern. In the new version of the paper to further clarify this we have added the following:
 - The definition of POE in the introduction section, sentence includes ‘parent-of-origin effects (POEs), which are non-additive genetic effects whose phenotypic influence depends on the parent-of-origin of alleles’ (lines 72-74).
 - Discussion of other possibilities that may introduce observed POE in sentences starting from ‘Other factors might also result in the increased full-sibling similarity observed in these CpGs’ and ‘In addition, POE could be introduced by mechanisms other than imprinting’ (lines 783-794).
 - As discussed before, we stress the point that our confidence in POE identified for CpGs is different for different categories of CpGs listed in Table 3 and discussion sections (lines 776-783).
- 4) For the concern that differential methylation is introduced by nearby reciprocal genotype, we note that first this won’t introduce the POE pattern we observed, both the imbalanced phenotypic similarity between pairs of nuclear family members with the same genetic similarity as revealed in VC analysis and the differential phenotypic level of individuals with reciprocal genotypes as revealed in POE-mQTL analysis, and we have excluded the influences of differential methylation by nearby reciprocal genotype by excluding CpGs that overlapped with SNPs in a preliminary QC step (supplementary text lines 22-23).

Q: You overlap your methylated regions with another study (ref 18) that tested for ASM and you use the overlap to validate you are identifying imprinting. However – this other study looked at multiple cell types (including adipose and skeletal muscle) did you

verify that the overlaps you find are using the same cell types? What percentage of their bisulfate converted bases needed to overlap your methylated regions to call it an overlap?

A: Regarding cell type, we don't think complete cell type match is necessary here, as the same imprinted regions are usually identified in multiple tissues, and even when there is a difference across tissues for a given region, seeing an imbalanced methylation between chromosomes shows the region is targeted by an imprinting mechanism. Given the limited number of genome-wide studies analysing imprinting in different tissues and the limited density of the examined regions we think it is relevant to describe regions that are potentially imprinted.

Cheung, W.A., et al. [15] identified CpGs that are significant in GIT (genotype-independent test) and they call the regions where those CpGs are located "non-genetically regulated imbalanced methylation regions". This is a list of genomic regions with at least 15 consecutive CpGs and where all CpGs did not show significant genetic methylation (ASM and mQTL $q \geq 0.1$) but showed significant imbalanced allelic methylation (GIT $q < 1 \times 10^{-5}$), and where the median imbalanced allelic methylation was highly significant ($\log_e(q) < -10$). When our enrichment analysis was performed, we annotated SNPs or CpGs to those regions and checked whether a POE-mQTL SNP or a POE-influenced CpG is more likely to locate in these non-genetically regulated imbalanced methylation regions than expected by chance only. As mentioned before, POE acts as an allelic effect by masking the allelic effect depending on parent of origin of that allele. If we assume that at least some of the 'non-genetic regulated imbalanced methylation regions' are indeed imprinted regions, enrichment of our POE-mQTL SNPs is expected in those regions. We added a new set of results (see lines 497-501) where we have calculated Fisher exact tests for both cis- and trans- POE-mQTL SNPs to test for enrichment in non-genetic regulated imbalanced methylation regions as reported in [19]. And more discussions in lines 704-710

We also observed enrichment of identified POE-influenced CpGs in these non-genetic regulated imbalanced methylation regions, however, when looking at the Figure 1, we could infer that in different scenarios of POE, if looking at the target CpGs of POE rather than the POE-mQTL SNPs, it is possible that any of the three ASM, mQTL and GIT tests become significant in regions containing those CpGs, therefore, the interpretation of the enrichment of POE-influenced CpGs in those regions is difficult, so we exclude that result in this version of the paper. We also delete the statements such as 'N=XX of the POE-influenced CpGs showed imbalanced allelic methylation between two paired chromosomes'.

Q: The assumption that differential methylation associated with a SNP is a consequence of POE rather than a driver does not seem valid and is not tested. Please clarify the logic.

A: As described before, the POEs targeted in this study are the genetic effects (detected using similarity between individuals) on CpGs masked by genomic imprinting. Those POEs can only be detected, if the genetic variance resulting from those genetic effects introduces inter-individual methylation variation of the target CpGs in the population. If there is no genetic variation, there is no phenotypic variation introduced by genetic variation. Although many studies have suggested that differential DNA methylation (note that in this context the

‘differential’ here means imbalanced DNA methylation level between two parental chromosomes, not difference of average DNA methylation level for a given CpG between individuals) is the underlying mechanism which establishes the initial ‘silencing’ effect in the imprinted regions, this ‘strong’ differential DNA methylation would not on its own result in inter-individual variation in a normal population such as GS:SFHS (as all individuals would be the same, each having a pair of differentially methylated parental chromosomes). As discussed previously, the most likely way that these ‘strong’ between-chromosome differential DNA methylation regions are associated with inter-individual methylation variation in a population is when there is a regulatory SNP located in one of those regions, so that the SNP’s genetic effect is masked by the imprinting and the ‘masked’ genetic variance introduces inter-individual variation at the methylation level of the regulated CpGs. Looking at the model in Figure 1, the initial strong ‘differential methylation introduces genomic imprinting thus the masking effect (the cross in black), but it is the SNPs whose genetic effect is masked that introduce variation in the targeted CpGs, and this downstream effect is what we target in this study (in the paper we do mention that POE-mQTL (where the black cross is located) are likely to be located in the initial imprinted regions, in other words, the ‘strong’ differential DNA methylation regions). In short we, and other literature[13] think this is how it works:

Imprinting (possibly introduced by differential DNA methylation between parental chromosomes)+SNPs => inter-individual variation introduced by POE on regulated CpGs

It is very unlikely that the causal direction is the reverse, as the imprinting mechanisms are established in early development stages regardless of the genotypes and the genotype is unchanged since an individual is conceived. To clarify these we have added figure 1 and more description in the new version of the paper in lines 80-82, 482-484,704-710.

Q: What does whole blood have to do with the phenotypes you are mapping? You are assuming whole blood methylation status is a good proxy for POE on the phenotypes you test. Please justify.

A: The association between CpGs in blood and BMI/obesity has been identified in a number of studies [16, 17]. We are not assuming whole blood status is a good proxy for POE on the phenotypes, we simply profile the CpGs influenced by POEs and test the phenotypic consequences of variation in those CpGs, we tried to clarify this in the text (Aim 4 in Table 1, lines 579-580). CpG status can be a good proxy for the phenotypes but that does not necessarily mean that POE on CpG is a good proxy for POE on phenotypes. The variation in phenotypes is determined by many factors, the POE signal on phenotypes is expected to be much weaker than POE signals in methylation as these phenotypes are, in principle, the result of more processes than just methylation (Note that in common with other intermediate metabolism traits such as gene expression and proteomic variation, fewer loci with larger effects are associated with methylomic variation than is generally the case for variation in phenotypic traits and diseases). Elucidating the molecular mechanisms through which POEs on CpGs are translated to effects on phenotypes (through regulation of expression and so on) is beyond the scope of this paper.

Q: You corrected for cell counts of some different cell types (immune cells) but there are many different cell types in whole blood and different cell types are known to have differing methylation profiles. Did you test the alpha diversity of the samples? Perhaps the genotypic diversity is actually driving cell type variation (and not methylation) and this is what you are detecting in most of your associations? Parent of origin effects are likely only a subset of the effects that you detect ... and likely a much smaller subset than claimed in the manuscript.

A: to answer your questions,

In our analysis, cell counts for major cells types were corrected before any downstream analysis was performed. This is also the procedure that many published studies used in the same tissue (whole blood) as ours and other tissues[18-20].

. The influence on methylation of the variation of major cell type (driven by genotypes or other factors) should be already removed and not influence the downstream analysis we performed, so we are quite confident on those factors not biasing our results. We do not believe that it is possible that any residual diversity in cell types, could generate the results we observe from the variance component or POE-mQTL analyses.

Both reviewer 1 and 2 are concerned about POE both being the only explanation for the effects we detected. As we detailed in the response to reviewer 1, we are quite confident that there is compelling evidence pointing that POE are the best explanation for most of our results, however it is clear that was not very easily understandable in the previous version of the manuscript. We tried to address that and make clearer the following points:

- 1) 586 of the 984 CpGs significant in our variance component analysis were also significant for the POE-mQTL analysis;
- 2) the reviewed version of the manuscript now includes a comparison with a recently published independent study ALSPAC and a replication rate of 65% for the POE-CpGs reported by ALSPAC using a POE-mQTL analysis is obtained in our variance component analysis(see lines 546-558);
- 3) additionally, in the updated manuscript we tried to clarify the two-stage analysis procedure we perform in Table 1 and lines 204-209; and highlight the different strength of evidence for POE in Table 3;
- 4) we added enrichment and association analyses separately for the candidate CpG group with POE-mQTL detected and the candidate CpG group without POE-mQTL detected to characterise the two groups (lines 560-575, 582-598);
- 5) we excluded the effect from shared environment between family members (F) (supplementary table s19 and dominance effect for majority of candidate CpGs (lines 786-790) for simplicity;
- 6) we highlighted in the discussion (lines 778-783) that the candidate CpGs without POE-mQTL detected should be treated with caution and need future validation.

Q: You do a lot of overlapping of your regions with regions reported in other data sets, but you do not test whether the SNPs or mQTL you identify fall in known imprinted control regions or CpG islands.

A: Yes, POE-mQTL SNPs are highly enriched in known imprinted regions. We add this analysis in the results section (lines 497-501).

Q: The argument that you are indeed detecting imprinted POE would be much more valid if you homed in on some resolved imprinting control regions (for example PEG3 or IGF2) as proof of principal. Then test to determine if you see the same patterns as you see in the novel sites you detect.

A: We detected POE-influenced CpGs in both PEG3, IGF2 and many other known imprinted genes (234 out of 984 candidate CpGs are located within 2kb distance of known imprinted regions. Details for each CpG see Table s3). We highlighted these in the updated version of the manuscript (lines 457-461) and added Figure s4 describing the pattern of POE in POE-influenced CpGs located in IGF2 and PEG3. We also added a new enrichment analysis using a tighter definition of known-imprinted regions (using a 2KB extension of the gene boundary in contrast to the 2MB as used in the previous version of the paper) (lines 459-461).

Q: Please justify the different methods of multiple tests correction you do at different stages of analysis.

A: The EPIC array DNA methylation data contains many correlated CpG sites [1] One of the consequences of this type of structure in the data is that the number of independent tests is lower than the total number of test performed. When performing genome-wide analysis, FDR is the most widely used method of multiple testing correction for gene expression analysis which shows similar correlation structures to methylation data [2]. FDR is also used by a number of studies for DNA methylation [3-5], together with Bonferroni correction. The choice of method should depend on the purpose of the analysis. Bonferroni is a very stringent correction that could produce a larger type II error rates, whereas the FDR method is concerned with the proportion of type I errors (usually 0.05) among the tested set, and is more powerful (thus has a lower type II error). It has been suggested that the Bonferroni method is too stringent for exploratory genomic analysis, particularly when validation analyses follow [6]. In our study, we perform a two-stage procedure for the identification of POE on methylation: the variance component analysis as the first stage, used for candidate identification, and the POE-mQTL analysis as the 2nd stage, which provides additional evidence for the candidates identified in stage 1. Therefore, FDR is used in the exploratory variance component analyses to decide which sites are candidates to follow up. FDR is also used in the second stage of analysis(POE-mQTL) following published mQTL studies [7, 8], particularly, a permutation based FDR, which accounts for the correlation structure of the data [8]. If we used a Bonferroni correction we would miss a large proportion of the 586 of the 984 CpGs significant in the VC. In this version of the manuscript we refer to an independent study (ALSPAC) that identified POE-influenced CpGs identified solely through POE-mQTL showing a replication rate of 65% ($N_{cpg}=129$), when compared to our

results suggesting that FDR is performing appropriately (this result is newly added in this version of the paper in lines 546-558).

Q: Why did you perform a Ward test and not a more standard LRT in your initial mixed model?

A: Because performing LRT in full model (GKFSC) is quite computationally intensive, particularly when the purpose is to test the significance of each random effect. The Wald test is much faster and provides a good approximation for significance of the components.

Q: You tried to replicate a very complex context-dependent finding (women >47) and it didn't replicate. This isn't a surprise as you need to subset the data very specifically – do you have any general, non-context dependent loci you could try to replicate? A general effect is not going to have the sample size issues you suggest underlie your lack of replication.

A: The finding itself is identified through a context-independent analysis (PheWAS is not context-dependent). After we identified this SNP through PheWAS, we further explored this association and observed that the association signal is strongest in females older than 47. This is the only significant association that passed the multi-trait association threshold in PheWAS results. As the reviewer suggests this is not surprising, the sample size is very limited compared to that of recent GWASs, particularly considering the model complexity in our analysis.

Q: The association of BMI with older women (who are likely peri- or post-menopausal and will have a greater BMI) is not a surprise given that the >47 year old cohort would be enriched for peri- and postmenopausal women. You should perform this analysis correcting for peri- or post-menopausal state to determine if your association holds, if it is truly with BMI, or if it is an artifact of another physiological process.

A: The association reported in the manuscript is the association between BMI and a SNP. This association is stronger in the >47 female group. As the reviewer points out, it would be possible that some physical post-menopausal changes are involved, but this fact would not invalidate the association itself, it would rather provide a mechanistic explanation for it. Unfortunately, due to the lack of menopausal information in GS:SFHS, we cannot test the hypothesis of menopause mediation in that sample. In UK Biobank, menopausal status was included, and we tested the association between BMI and the target SNP in pre- and post-menopausal groups separately, here is the result:

	Est(POE)	err	t	Pval	N	Ref	Alt
Had menopause female	-0.02697	0.042357	-0.637	0.5294	99	C	G
Not had menopause female	-0.00444	0.004535	-0.98	0.32735	2216	C	G

The results are consistent with what we observed before in both GS:SFHS and UK Biobank, in the older female group ('had menopause') the SNP has a larger effect size estimate of POE compared with the 'has not had menopause'. However, the association is still not significant. These results are not included in the paper.

Q: How many context dependent associations did you find and how many were 'general' associations across the entire population tested?

A: As mentioned in another comment before, the results presented in the paper are non-hypothesis driven and non-context dependent genome-wide scans to identify POE-influenced CpGs and their regulatory SNPs. Also non-hypothesis driven and non-context dependent scans to identify phenotypic consequences of variation in identified POE-influenced CpGs and their associated SNPs. The study design is shown in table 1 and described in lines 202-211. The only results involving specific age/sex context are those exploring the lack of replication for significant SNP identified in the PheWAS (only one SNP).

Q: The text should be edited for brevity. A lot of the language in the manuscript is exactly the same as that in the supplement and a lot of modifying words are unnecessary and should be completely eliminated.

A: We have reduced overlap as much as possible and removed many modifying words.

Q: "Given the fact that there are multiple possible imprinting inheritance patterns [34], each of which is expected to display a different covariance structure between parents and offspring and between full-siblings (Figure 1)." What does this sentence mean?

A: We have modified the sentence to 'There are several possible imprinting inheritance patterns [21], each of which is expected to produce a characteristic covariance structure between parents and offspring and between full-siblings (see Figure 2 for examples). ' in lines 283-285 (note that the original Figure 1 is Figure 2 in this new version).

Q: Figure 1 legend should say "expected" covariance structures rather than "corresponding".

A: Edited as suggested.

Reviewer #3 (Remarks to the Author):

The authors use a well defined and well studied Scottish cohort where phenotypes and DNA methylation data have been collected. Specifically there are 1,668 offspring in the cohort that have both parent-of-origin assigned to alleles and DNA-methylation data that were used in the POE-mQTL analysis. The authors modelled parent of origin

effects using phenotypes, variants and EPIC data with a model choice that offered maternal, paternal or complex classification of effects.

The authors define imprinting and P-O-O effects and note that there are only a few imprinted gene in the genome but that parental affects can be the result of regulatory variants around the genome and go on to show this in the data analysis.

The analysis looked for CpGs potentially influenced by POEs, then complex traits associated with DNA methylation levels.

Q: The parameters of the study subjects (collection day and so forth), appear multiple times, this could be parred back a bit to reduce repetition.

A: We have reduced repetition where possible.

Q: The authors talk about assigning CpGs to regions of “different chromatin and histone modification states”, this will require a bit more explanation and rationale, other than you can do it because there are data in the public domain. Consideration for matched tissue is not mentioned, why this is useful is not discussed.

A: As suggested, in the new version we added sentences including ‘DNA methylation dynamics is associated with altered chromatin structure [45], and coupled with histone modifications in relevant tissues [46], and transcription factor binding sites.’, and ‘A lymphoblastoid cell line (GM12878) and an immortalised myelogenous leukaemia cell line (K562) were used in this annotation as they are the two cells produced from blood among primary cell lines with abundant annotation information in the ENCODE project [22].’ in lines 370-375

Not surprisingly, the study found genetics as the largest factor in methylation variation. The “shared environmental effects” measured were to do with living in the same place as family members of partners.

Q: Does a genetic variant ‘regulate a CpG in a parent of origin fashion’? Could the authors use a more mechanistic way of terming this relationship?

A: Thanks for this suggestion. In the new version we rephrased it as ‘genetic variants (POE-mQTLs) that regulate the CpGs through POE’ (line 665)

Q: “The identified CpGs influenced by POEs were enriched in polycomb repressed regions and ncRNA”. The CpGs would be in REGIONS enriched in polycomb, rather than the CpG site itself.

A: Thanks for suggesting this. We have now rephrased this including the word regions throughout the manuscript.

Q: “These CpGs also enriched in genes”?? doesn’t make sense.

A: Thanks for pointing this out. We have rephrased this as ‘the 984 CpGs were also enriched in genic regions of genes containing methylation sites associated with body mass index (lines 466-467).

Q: “PIEZO1P2 and its regulatory regions should be treated as targets of obesity-related Research” or maybe targets for post-menopausal (eg hormonal) differences compared to pre-47 year old females?

A: We agree, we also considered the possible role that menopause plays here. It is possible that some physical changes post-menopausal are involved in this association, but due to the lack of menopausal information in GS:SFHS, we cannot test the influence of menopause in that sample. In UK Biobank, menopausal status was measured, and we test the association between BMI and the target SNP in pre- and post-menopausal groups separately. The table below shows the result:

	Est(POE)	err	t	Pval	N	Ref	Alt
Had menopause female	-0.02697	0.042357	-0.637	0.5294	99	C	G
Not had menopause female	-0.00444	0.004535	-0.98	0.32735	2216	C	G

This is consistent with what we observed before, in the older female group (‘had menopause’) the SNP has a larger effect size estimate of POE compared with the ‘has not had menopause’. However, as before, the association is not significant.

Overall the study makes use of a strongly curated cohort and applies an epigenome style analysis to the genetic and phenotypic data to model parent of origin effects and identify regions of interest. The study places the effects into categories and looks to explain effects in a more consistent way that some others have done (which it cites). The study also highlights a waist size phenotype which is one of the main findings of the paper.

References

1. Lovkvist, C., et al., *DNA methylation in human epigenomes depends on local topology of CpG sites*. Nucleic Acids Research, 2016. **44**(11): p. 5123-5132.
2. Cheng, C. and S. Pounds, *False discovery rate paradigms for statistical analyses of microarray gene expression data*. Bioinformatics, 2007. **1**(10): p. 436-46.
3. Lister, R., et al., *Human DNA methylomes at base resolution show widespread epigenomic differences*. Nature, 2009. **462**(7271): p. 315-322.
4. Wilson, L.E., et al., *An epigenome-wide study of body mass index and DNA methylation in blood using participants from the Sister Study cohort*. International Journal of Obesity, 2017. **41**(1): p. 194-199.
5. Marioni, R.E., et al., *Meta-analysis of epigenome-wide association studies of cognitive abilities*. Mol Psychiatry, 2018.
6. Bonder, M.J., et al., *Disease variants alter transcription factor levels and methylation of their binding sites*. Nature Genetics, 2017. **49**(1): p. 131-138.
7. Westra, H.J., et al., *Systematic identification of trans eQTLs as putative drivers of known disease associations*. Nature Genetics, 2013. **45**(10): p. 1238-U195.

8. Namjou, B., et al., *Phenome-wide association study (PheWAS) in EMR-linked pediatric cohorts, genetically links PLCL1 to speech language development and IL5-IL13 to Eosinophilic Esophagitis*. *Front Genet*, 2014. **5**: p. 401.
9. Salnikova, L.E., M.B. Khadzhieva, and D.S. Kolobkov, *Biological findings from the PheWAS catalog: focus on connective tissue-related disorders (pelvic floor dysfunction, abdominal hernia, varicose veins and hemorrhoids)*. *Hum Genet*, 2016. **135**(7): p. 779-95.
10. Barnado, A., et al., *Phenome-wide association study identifies marked increased in burden of comorbidities in African Americans with systemic lupus erythematosus*. *Arthritis Res Ther*, 2018. **20**(1): p. 69.
11. Reik, W. and J. Walter, *Genomic imprinting: parental influence on the genome*. *Nat Rev Genet*, 2001. **2**(1): p. 21-32.
12. Cuellar Partida, G., et al., *Genome-wide survey of parent-of-origin effects on DNA methylation identifies candidate imprinted loci in humans*. *Hum Mol Genet*, 2018.
13. Lawson, H.A., J.M. Cheverud, and J.B. Wolf, *Genomic imprinting and parent-of-origin effects on complex traits*. *Nature Reviews Genetics*, 2013. **14**(9): p. 608-617.
14. Wolf, J.B., et al., *Genome-wide analysis reveals a complex pattern of genomic imprinting in mice*. *PLoS Genet*, 2008. **4**(6): p. e1000091.
15. Cheung, W.A., et al., *Functional variation in allelic methylomes underscores a strong genetic contribution and reveals novel epigenetic alterations in the human epigenome*. *Genome Biol*, 2017. **18**(1): p. 50.
16. Dick, K.J., et al., *DNA methylation and body-mass index: a genome-wide analysis*. *Lancet*, 2014. **383**(9933): p. 1990-8.
17. Wahl, S., et al., *Epigenome-wide association study of body mass index, and the adverse outcomes of adiposity*. *Nature*, 2017. **541**(7635): p. 81-86.
18. McRae, A.F., et al., *Contribution of genetic variation to transgenerational inheritance of DNA methylation*. *Genome Biol*, 2014. **15**(5): p. R73.
19. McCartney, D.L., et al., *Altered DNA methylation associated with a translocation linked to major mental illness*. *NPJ Schizophr*, 2018. **4**(1): p. 5.
20. Richmond, R.C., et al., *DNA methylation and body mass index: investigating identified methylation sites at HIF3A in a causal framework*. *Diabetes*, 2016: p. db150996.
21. Lawson, H.A., J.M. Cheverud, and J.B. Wolf, *Genomic imprinting and parent-of-origin effects on complex traits*. *Nat Rev Genet*, 2013. **14**(9): p. 609-17.
22. Consortium, E.P., *The ENCODE (ENCyclopedia Of DNA Elements) Project*. *Science*, 2004. **306**(5696): p. 636-40.

Reviewer #1 (Remarks to the Author):

The authors have responded to comments well, and in a lot of detail.

My concerns remain about how large some of the p-values that pass the FDR in the first stage are. Using FDR in the first stage is justified by "we use this first stage to reduce the number of probes ... that we take for the second stage of the analyses", and such an approach would be fine. However, this is not just used for number reduction, with the abstract clearly presenting these as novel results: "The scan identified 733 independent (984 total) methylation CpGs potentially influenced by parent-of-origin effects of which 331 had not previously been identified".

The overlap with the now published Cuellar Partida study provides more confidence in the results. However, a high percentage replication of the Cuellar Partida does not imply that the false positive rate in this study is well controlled. The better statistic would be the proportion of this studies results that are replicated by Cuellar Partida. I note that the publication of the Cuellar Partida study does not alter the novelty of this study in my opinion.

Finally, it appears that my comment on the QQplot for POE-PheWAS (Figure S6) was misinterpreted. The four most significant results from this study are all less significant than expected by chance. This does not "suggested that the test is over-conservative", but more directly suggests that there are no significant results.

Reviewer #2 (Remarks to the Author):

The revised manuscript by Zeng et al. is much improved with the addition of the VC analysis and enrichment analyses. A couple additional questions:

Can you please explain the designation of random and fixed effects in your mixed models? In text S2, why are clinic, batch, year, position, and day considered fixed effects? It seems these should be random effects in this model along with date and ID (i.e. there is no reason to believe these variables should have any global effects on variation and should therefore be random). Likewise, in the main text, for the VCA, shouldn't season, time and day be random effects for the same reason?

How were the overlaps of your regions with regions from ref 13 performed? What percentage of bases in a region were required to overlap? In the regions that did overlap, how many of the differentially methylated regions showed the same POE pattern?

In response to previous questions from reviewers, you explain that you are detecting parent-of-origin effects that interact with genotypic variation to result in differential methylation that in some cases associate with phenotype. (POExSNP -> DMeth -> Phenotype). But then when responding to the question 'what does whole blood have to do with the phenotype?' you state that parent-of-origin effects on methylation is not predictive of POE on phenotype. Please explain.

In response to the question of whether or not your association is an artifact of a cohort of women likely to be peri- and postmenopausal you test the association in an independent cohort by separating women who have and who have not had menopause. The POE is stronger in the women who have had menopause but the error is much, much higher. Why not simply test a model including both POE and menopausal state and ask how much variance in BMI POE explains and how much menopause explains?

The figures are much improved as are the figure legends.

Reviewer #3 (Remarks to the Author):

The authors have batted back many of the comments from the reviewers, the one aspect they failed to deal with is the use of terminology which does not take into account the known mechanistic facts about imprinting.

The first two sentences of the abstract are uncomfortable, especially the statement "Most studies on imprinting caused by parent-of-origin effects...". Really ignoring the dogma to date. These two sentences need to be removed.

The use of parent of origin effects is much better than using the term Imprinting in some of the contexts as previously.

"Several scenarios can lead to POEs, but the most common is genomic imprinting" is the term "scenarios" really the best choice?

It's surprising that ICRs are not defined and used to explain some of the areas of difficulty experienced by the authors in making some of the material clearer.

Specific queries from me have been argued or addressed.

We thank the reviewers for their comments and apologise that our previous version was unclear in places. Below we show reviewers' comments (**in bold**) and our responses (in plain text).

Reviewer #1 (Remarks to the Author):

Q:The authors have responded to comments well, and in a lot of detail.

My concerns remain about how large some of the p-values that pass the FDR in the first stage are. Using FDR in the first stage is justified by "we use this first stage to reduce the number of probes ... that we take for the second stage of the analyses", and such an approach would be fine. However, this is not just used for number reduction, with the abstract clearly presenting these as novel results: "The scan identified 733 independent (984 total) methylation CpGs potentially influenced by parent-of-origin effects of which 331 had not previously been identified".

The overlap with the now published Cuellar Partida study provides more confidence in the results. However, a high percentage replication of the Cuellar Partida does not imply that the false positive rate in this study is well controlled. The better statistic would be the proportion of this studies results that are replicated by Cuellar Partida. I note that the publication of the Cuellar Partida study does not alter the novelty of this study in my opinion.

A: In response to the reviewer's concerns we have redrawn table 3 to clarify for how many of the 984 sites we have detected we have strong evidence from our analyses and how many are clearly novel. We classify as having 'strong evidence' those 586 sites both detected by our variance component analyses and which have a POE-mQTL. We classify the remaining 398 sites as having moderate evidence. For each of these two classifications we break them down according to whether they are replicated by a previous detected imprinted region (within 2kb of that region), whether they are novel (more than 2Mb away from a previously detected imprinted region) or between 2kb and 2Mb of a previously detected region (i.e., it is unclear whether they are a replicate or a novel site). This results in our detection of 191 sites for which we have strong evidence and which are novel and when accounting for correlation between these sites, they represent 171 independent sites. It is these two latter numbers (191 and 171) that we now quote in the abstract. Unfortunately we cannot provide the proportion of our study's results that is significant at a replication threshold in the results of Cuellar Partida as this latter study only reports genome-wide significant results and in any event uses a different array with fewer CpGs measured than used here. In fact we believe that a proportion of the remaining sites will also be new discoveries that will be validated in future. We now also highlight in the abstract that the 733 loci were the independent CpGs that passed our FDR correction threshold and give this threshold in the abstract.

Q: Finally, it appears that my comment on the QQplot for POE-PheWAS (Figure S6) was misinterpreted. The four most significant results from this study are all less significant than expected by chance. This does not "suggested that the test is over-conservative", but more directly suggests that there are no significant results.

A: We note the comment from the reviewer about Figure 6, but notwithstanding this comment the trait associations we report are above the stringent significance threshold we use and so formally significant and hence we believe these results should be reported.

Reviewer #2 (Remarks to the Author):

Q: The revised manuscript by Zeng et al. is much improved with the addition of the VC analysis and enrichment analyses. A couple additional questions:

Can you please explain the designation of random and fixed effects in your mixed models? In text S2, why are clinic, batch, year, position, and day considered fixed effects? It seems these should be random effects in this model along with date and ID (i.e. there is no reason to believe these variables should have any global effects on variation and should therefore be random). Likewise, in the main text, for the VCA, shouldn't season, time and day be random effects for the same reason?

A: In performing these analyses we wanted to avoid as much as possible spurious effects due to potential confounding with nuisance effects. For potential confounders with few levels such as clinic, batch, etc. including these as fixed effects maximises the variance they explain and it is possible to take this cautious approach even though it may result in the removal of true biological signal. For factors with large numbers of levels, it is only possible to include them as random effects and this still reduces the risk of spurious effects whilst not completely ablating any true signal.

Q: How were the overlaps of your regions with regions from ref 13 performed? What percentage of bases in a region were required to overlap? In the regions that did overlap, how many of the differentially methylated regions showed the same POE pattern?

A: Reference 13 (Cuellar Partida et al.) reports precise CpGs and SNPs and when we consider overlap with this study it is precise overlap. Note however when we claim novelty for identified CpGs influenced by POEs (e.g. in Table 3) we use a conservative 2Mb distance (4 Mb window) to be sure that there is no overlap with previously identified sites.

Q: In response to previous questions from reviewers, you explain that you are detecting parent-of-origin effects that interact with genotypic variation to result in differential methylation that in some cases associate with phenotype. (POExSNP -> DMeth -> Phenotype). But then when responding to the question 'what does whole blood have to do with the phenotype?' you state that parent-of-origin effects on methylation is not predictive of POE on phenotype. Please explain.

A: In considering the potential effect of POE effects on CpGs (or indeed any cellular phenotype such as gene expression or the proteome) there is often no single ideal tissue or cell type that can reflect the complexity of genetic and environmental signals on the trait. Many studies of the genetics of gene expression or protein levels have shown that blood based measures may be associated with a range of different diseases (e.g. <http://www.nature.com/articles/s41586-018-0175-2>). Additionally, for a surprising number of traits, including those related to obesity, blood based methylation measures have been associated with trait variation and other studies suggest that these signals may in part be independent of additive genetic effects (Shah et al. Am J Hum Genet. 2015 Jul 2;97(1):75-85. doi: 10.1016/j.ajhg.2015.05.014). In addition, any influence of individual CpGs will be blurred by the influence of all other genetic and environmental factors influencing a trait. Thus even for a CpG that influences (or is influenced by) trait variation, a POE on that CpG may not be observed as a clear POE at the level of the trait.

Q: In response to the question of whether or not your association is an artifact of a cohort of women likely to be peri- and postmenopausal you test the association in an independent cohort by separating women who have and who have not had menopause. The POE is stronger in the

women who have had menopause but the error is much, much higher. Why not simply test a model including both POE and menopausal state and ask how much variance in BMI POE explains and how much menopause explains?

A: In looking at the POE effect on BMI the standard error for POE in women who have had menopause is higher because the number of individuals in this group is very small. That is why although we saw a consistently higher point estimate of effect size of POE in this group as in the group of women in the >47 years old category, we still do not have enough power to draw a statistically solid conclusions on the replication in this particular case. Rather than the variance explained by POE or menopause in BMI, the effects we detected are more relevant to the interaction effect between POE and menopause (or age), but again as the results show, the power is not enough for a solid replication. We thought the results of this exploratory analysis were still worth reporting, bearing in mind the caveats.

The figures are much improved as are the figure legends.

Reviewer #3 (Remarks to the Author):

Q: The authors have batted back many of the comments from the reviewers, the one aspect they failed to deal with is the use of terminology which does not take into account the known mechanistic facts about imprinting.

A: We have attempted to take further account of the known mechanisms when revising the manuscript. We have always considered known ICRs when exploring overlap of the effects we detected with known imprinted regions and we now emphasise this explicitly. We have also looked explicitly at the overlap between the sites we identify and ICRs and find that the enrichment of sites we have detected within ICRs (OR for *cis* POE-mQTL SNPs = 2.4, OR for *trans* POE-mQTL SNPs = 1.3) is less than with known imprinted regions (that includes ICR), i.e. OR for *cis* POE-mQTL SNPs = 7.8, OR for *trans* POE-mQTL SNPs = 10.2.

Q: The first two sentences of the abstract are uncomfortable, especially the statement “Most studies on imprinting caused by parent-of-origin effects...”. Really ignoring the dogma to date. These two sentences need to be removed. The use of parent of origin effects is much better than using the term Imprinting in some of the contexts as previously.

A: We have replaced the first two sentences of the abstract with this one:

“Parent-of-origin effects (POE) exist when there is differential expression of alleles inherited from the two parents, which influences DNA methylation, gene expression and phenotypes.”

In addition we have been careful about the use of “imprinting” rather than parent of origin effects throughout.

Q: “Several scenarios can lead to POEs, but the most common is genomic imprinting” is the term “scenarios” really the best choice?

A: We have rephrased this sentence to ‘There are several possible causes of observed POEs, but the most common is genomic imprinting.’

Q: It’s surprising that ICRs are not defined and used to explain some of the areas of difficulty experienced by the authors in making some of the material clearer.

A: As mentioned we have now included in the paper that the enrichment of sites we have detected within ICRs (OR for *cis* POE-mQTL SNPs = 2.4, OR for *trans* POE-mQTL SNPs = 1.3) is less than with known imprinted regions (that includes ICR), i.e. OR for *cis* POE-mQTL SNPs = 7.8, OR for *trans* POE-mQTL SNPs = 10.2.

Specific queries from me have been argued or addressed.

Reviewer #1 (Remarks to the Author):

All my comments have been addressed in the revised manuscript.

Reviewer #2 (Remarks to the Author):

The authors have addressed my concerns and the manuscript is much improved.